# Barriers and facilitators to anti-retroviral therapy adherence among adolescents aged 10 to 19 years living with HIV in sub-Saharan Africa: A mixed-methods systematic review and meta-analysis

Londiwe D. Hlophe[1,2], Jacques L. Tamuzi[1], Constance S. Shumba[3], Peter S. Nyasulu[1,4]*

1 Division of Epidemiology and Biostatistics, Faculty of Medicine and Health Sciences, Stellenbosch University, Cape Town, South Africa, 2 Department of Environmental Health Sciences, Faculty of Health Sciences, University of Eswatini, Mbabane, Kingdom of Eswatini, 3 School of Nursing and Midwifery, Aga Khan University, Nairobi, Kenya, 4 Division of Epidemiology and Biostatistics, School of Public Health, Faculty of Health Sciences, University of the Witwatersrand, Johannesburg, South Africa

* pnyasulu@sun.ac.za

## Abstract

### Background

Human Immunodeficiency Virus (HIV) significantly affects adolescents globally, with the sub-Saharan Africa (SSA) reporting a high burden of the disease. HIV testing, treatment, and retention to care are low among adolescents. We conducted a mixed-method systematic review to assess anti-retroviral therapy (ART) adherence; barriers and facilitators to ART adherence and ART outcomes among adolescents living with HIV and on ART in sub-Saharan Africa.

### Methods

We conducted searches in four scientific databases for studies conducted between 2010 and March 2022 to identify relevant primary studies. Studies were screened against inclusion criteria and assessed for quality, and data was extracted. Meta-analysis of rates and odd ratios was used to plot the quantitative studies and meta-synthesis summarized the evidence from qualitative studies.

### Results

A total of 10 431 studies were identified and screened against the inclusion/ exclusion criteria. Sixty-six studies met the inclusion criteria (41 quantitative, 16 qualitative, and 9 mixed-methods study designs). Fifty-three thousand two hundred and seventeen (53 217) adolescents (52 319 in quantitative studies and 899 in qualitative studies) were included in the review. Thirteen support focused interventions for improved ART adherence were identified from quantitative studies. The plotted results from the meta-analysis found an ART

**Data Availability Statement:** All relevant data are within the manuscript and its Supporting Information files.

**Funding:** The authors received no specific funding for this work.

**Competing interests:** The authors have declared that no competing interests exist.

**Abbreviations:** HIV, Human immunodeficiency virus; AIDS, Acquired immunodeficiency syndrome; ALHIV, adolescents living with HIV; ART, antiretroviral therapy; CD4, cluster of differentiation 4; RNA, Ribonucleic acid; COVID-19, Corona virus disease 19; COREQ, Consolidated Qualitative Study; ENTREQ, Enhancing Transparency in Reporting the synthesis of Qualitative research; PROSPERO, Prospective register of systematic reviews; MMAT, Mixed Methods Appraisal tool; PRISMA, Preferred Reporting Items for Systematic Reviews and Meta-Analyses; SSA: sub-Saharan Africa; STROBE, Strengthening the Reporting of Epidemiological Studies; UNAIDS: The Joint United Nations Programme on HIV/AIDS; WHO, World Health Organization; CINAHIL, Cumulative index of nursing and allied health literature; PubMed, Public/ publisher Medline; LTFU, Loss to follow up; VLS, Viral load suppression; SMS, Short messages services; TB, Tuberculosis; IeDEA, International epidemiology database to evaluate AIDS; VL: Viral load.

adherence rate of 65% (95%CI 56–74), viral load suppression was 55% (95%CI 46–64), un-suppressed viral load rate of 41% (95%CI 32–50), and loss to follow up of 17% (95%CI 10–24) among adolescents. Meta-synthesis found six themes of barriers to ART (social, patient-based, economic, health system-based, therapy-based, and cultural barriers) in both the qualitative and quantitative studies, and three themes of facilitators to ART were also identified (social support, counselling, and ART education and secrecy or confidentiality) from qualitative studies.

## Conclusion

ART adherence remains low among adolescents in SSA despite multiple interventions implemented to improve ART adherence. The low adherence rate may hinder the attainment of the UNAIDS 2030 targets. Additionally, various barriers to ART adherence due to lack of support have been reported among this age group. However, interventions aimed at improving social support, educating, and counselling adolescents may improve and sustain ART adherence.

## Trial registration

**Systematic review registration:** PROSPERO CRD42021284891.

## 1. Background

Adolescents are the most affected age group globally, with 1.75 million (1.16–2.3 million) of adolescents living with HIV (ALHIV) and 90% of these adolescents are found in the sub-Saharan Africa (SSA) region by July 2021 [1]. In 2020, adolescents accounted for 11% of new infections and 5% of AIDS-related deaths globally [2]. HIV testing is low among adolescents, especially in SSA countries. In a study conducted by Asaolu et al., 26.5% of adolescents aged 15 to 19-years had ever tested for HIV in sub-Saharan Africa, thus the proportion of those living with HIV and on treatment is low [3]. HIV treatment outcomes such as viral load suppression and AIDS-related death rate are also poor among adolescents due to low retention to care and poor treatment adherence rate [4, 5]. Accordingly, viral load suppression is low among adolescents when compared to adult population [6].

Viral load suppression (VLS) defined as viral load <50 RNA copies per ml of blood is associated with optimum ART adherence [7]. Optimum ART adherence is defined as correctly taking of medication doses above 95% [8]. Among adolescents, ART adherence is reported to be low and even lower in the SSA region. The sub-optimum adherence (less than 95% of correctly taking medication) is as a result of a number of factors namely; social, economic, therapy-related and health system-related and patient-related barriers [9].

Several interventions have been implemented to counteract these barriers and thus facilitate ART adherence among people living with HIV. These interventions are mainly focused in improving social support through the use of reminders and peers [10, 11]. Additionally, interventions aimed at improving facility-based barriers such as ART clinics and combination therapy have been implemented yielding positive ART treatment outcomes [12, 13]. World Health Organization guidelines have enabled the improved diagnosis and treatment initiation of people living with HIV. These include the 2010 guidelines which broadened the eligibility to ART to CD4 count of $\leq 350$ cells/mm$^3$ for all regardless of the clinical stage [14].

Targets aimed at collective effort to improve clinical outcomes have also been set such as the fast-track commitments by UNAIDS. These are the 90% reduction of new HIV cases by the year 2030 and the 61% reduction in HIV prevalence between 2010 and 2050 [15, 16]. Additionally, the fast-track 2030 targets aiming at 95% of those living with HIV knowing their HIV status, 95% of them on ART and 95% of those on ART with suppressed viral load. However, adolescents in the SSA region present poor outcomes as 43% are initiated on ART treatment with only 31% of those initiated on ART continue to access ART. Among those continually accessing ART, 30% are viral load suppressed [17, 18].

The COVID-19 pandemic has further negatively influenced HIV outcomes globally [19, 20]. For instance, in 2018, viral load suppression among those on ART was 58% increasing to 72% in 2020 [21, 22]. In 2021, globally, there was a reduction in viral load suppression rate among those on ART globally as it was 66% from 72% in 2020 [23]. Even though adolescent data is scanty, studies comparing viral load suppression rate pre-COVID-19 and during the pandemic have generally indicated a decrease in VLS. For instance, a study conducted in Durban, South Africa in 2020 among ALHIV, VLS was 98.5% pre-COVID-19 and dropped to 83.6% during the COVID-19 period [24]. The reduction in VLS was also seen in a study conducted in Atlanta [25]. However, a study conducted in the Democratic Republic of Congo a study reported an increase in VLS during COVID-19 pandemic especially in rural [26]. The increase was a result of closeness to health facilities, the differentiated care model change from monthly refill to multi-months refill cutting transport costs and clinical factors [26]. Nevertheless, the general drop in VLS among ALHIV is an indication of poor ART adherence during the COVID-19 pandemic due to lockdowns [27–31].

As a result of the lockdowns, there were disruptions in access to HIV prevention and control services resulting in new infections and severe HIV treatment outcomes, especially among adolescents from SSA [24, 26, 32–34]. Within SSA, COVID-19 prevention and treatment options were limited with low vaccination coverage of 3% yet COVID-19 infection has been associated with severe outcomes among people living with HIV [1, 19–21]. Studies on impact of COVID-19 among adolescents by members of the Adolescent HIV Prevention and Treatment Implementation Science Alliance (AHISA) program revealed that the COVID-19 pandemic threatened the accessibility and availability of ART for adolescents [31, 35]. The lockdowns restricted access to health care facilities for refills, clinical support such as counselling session and other social support programs [27, 28, 35]. Secondly, healthcare workers were re-deployed in COVID-19 program thus reducing the number of staff members resulting to under-staffing [27, 29]. Thirdly, funding was re-directed to COVID-19 programs while HIV programs were affected resulting to adolescents' programs stalled [27, 29]. Furthermore, loss of income and restriction on travel due to COVID-19 lockdowns impacted on ART access and adherence as there were no transport and funds for transport costs to health facilities [27–29, 35]. In a study conducted in 2022 among ALHIV and on ART in South Africa revealed that COVID-19 lockdown restrictions have been associated with mental health problems among adolescents living with HIV as a result of the isolation, quarantine, closing of schools which reduced the support systems for adolescents especially from schools and peers [30, 33, 36, 37]. Lastly, COVID-19 prevention and control strategies have been associated with social isolation, loss to follow up and poor ART adherence due to food insecurity and poverty as a result of the COVID-19 pandemic [27, 32, 35].

Nevertheless, advances had been made pre-COVID-19 pandemic in the control and prevention of HIV yet adolescents still presented poor HIV treatment outcomes [18]. There is scanty comprehensive adolescent-focused data on the barriers, facilitators of ART adherence, and HIV treatment outcomes in SSA even pre-COVID-19 pandemic. Therefore, our review aimed at systematically and critically reviewing literature on ART adherence among

adolescents living with HIV (ALHIV) in SSA by exploring evidence on: (a) the barriers and facilitators of ART adherence; (b) ART outcomes (ART adherence rate, loss to follow up, viral load suppression and non-suppression rates); (c) Types of interventions aimed at improved ART adherence among ALHIV from SSA.

## 2. Methods

The systematic review is registered with PROSPERO (Registration: CRD42021284891) [38]. The results are reported according to the Preferred Reporting Items for Systematic Reviews and Meta-Analyses (PRISMA statement) [39], and the Enhancing Transparency in Reporting the synthesis of Qualitative research (ENTREQ) recommendations [40].

### 2.1 Search strategies

A literature search including only studies conducted from 2010 to March 2022 was conducted. The following databases were used for this literature search; PubMed, Cochrane Review, Scopus on Excerpta Medica Database (Embase) and CINAHL.

To identify studies, the following search terms were used: *(Antiretroviral OR ART) AND (adherence Or Compliance) AND (Adolescents) AND (sub-Saharan Africa\* OR sub-Saharan Africa region)*. Full search terms are found in S1 Table.

The inclusion criteria were research studies (qualitative, quantitative, and mixed method research studies) reporting on ART adherence outcomes among adolescents aged 10 to 19-years in sub-Saharan Africa. We only included English published articles. The following outcomes were included:

- ART adherence rate: proportion of adolescents with optimum ART adherence among total adolescent in the study

- ART adherence barriers: list and proportion of adolescents reporting either patient or individual, economic, social, health system-related, therapy-related and cultural barriers faced by participants in attaining optimum ART adherence [9, 41, 42].

- Lost to follow up (LTFU): proportion adolescents lost to follow up.

- Viral load (VL) suppression: proportion of adolescents acquiring VL suppression (VL <50 copies/ml) in the study.

- Unsuppressed VL: proportion of adolescents with VL>50 copies/ml.

### 2.2. Study selection

Two investigators (LDH and JLT) independently assessed studies citations against the inclusion criteria. For citations where there was eligibility disagreement, PSN further independently resolved the disagreements. LDH and JLT further independently assessed included studies for eligibility through reading through the full text and disagreements were resolved through consensus between the two investigators. The review process is attached as Fig 1.

### 2.3. Data extraction

We imported search outputs into Covidence library and removed duplicate references. Two data extraction forms were designed for quantitative and qualitative study designs. Mixed methods data extraction was conducted as separated in quantitative and qualitative designs. LDH and JLT independently extracted descriptive and analytical data from included studies (Tables 1 and 2). Extracted data included the following: Author (s), title, year of publication,

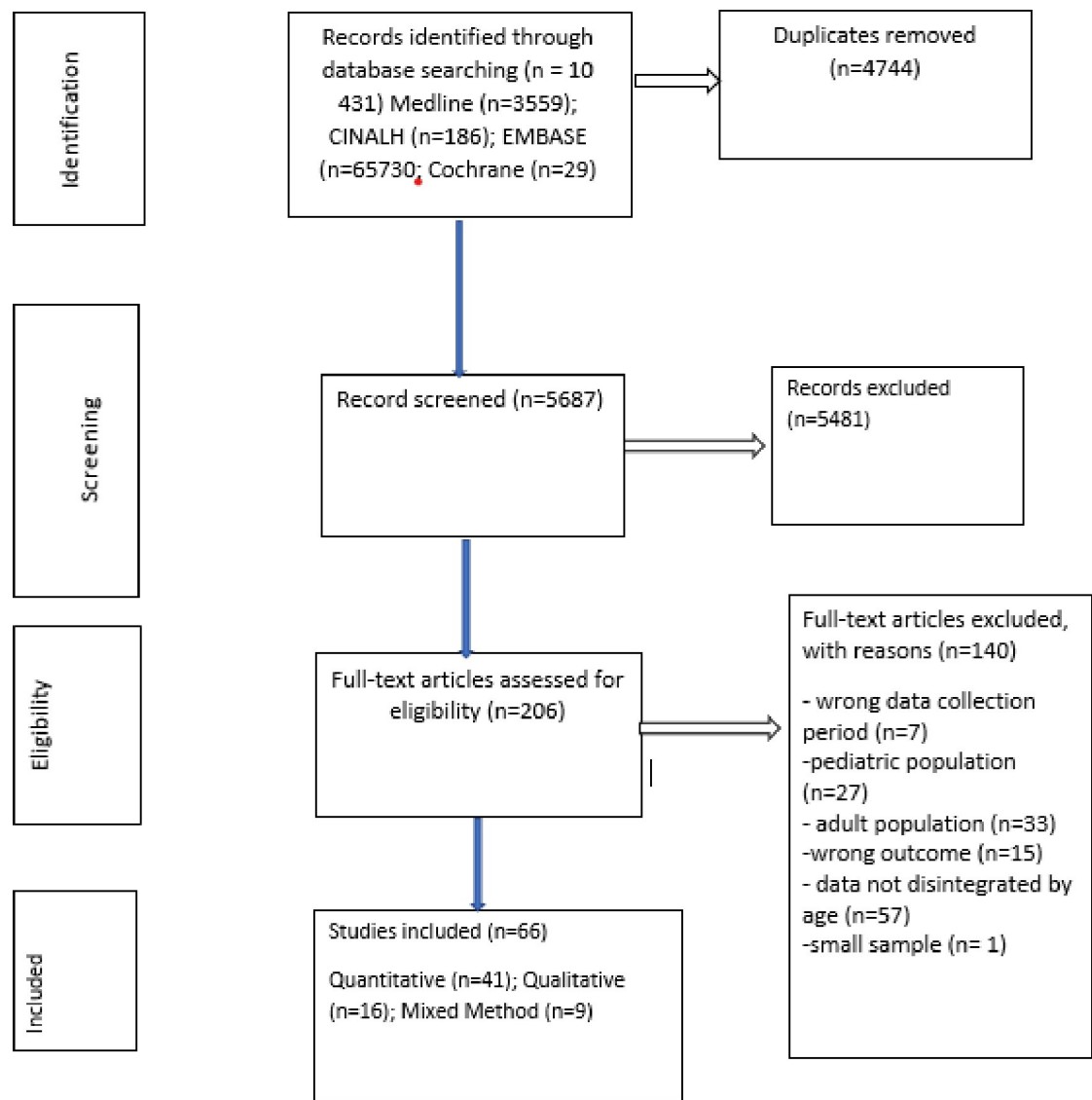

**Fig 1. Flow diagram of studies including barriers and facilitators to anti-retroviral therapy adherence among HIV infected adolescents aged 10 to 19 years living in sub-Saharan Africa.**

country, study design, study characteristics, participants demographic data, type of intervention, themes or outcomes, and final findings. Secondly, barriers to ART adherence were grouped into patient-related, therapy-related, health system-related, social, economic, and cultural factors (9) (Tables 3 and 4). Lastly, facilitators to ART adherence were extracted from each included qualitative study (Table 5). Consensus of data extraction was obtained through discussions between JLT and LDH and JLT finally went through all citations and data extracted to ensure accuracy. In case of any discrepancy, PSN was consulted. Upon consensus on qualified studies, LDH and JLT entered relevant data for the excel data extraction forms from potentially eligible studies. Strengthening the reporting of epidemiological studies (STROBE) and the Consolidated Qualitative Study (COREQ) Reporting Standards for both quantitative and qualitative studies have guided the design of these types of forms [43].

**Table 1. Overview of included qualitative studies.**

| Study ID and settings | Aim/s of the study | Participants/ Population | Recruiting from (Sampling strategy) | Data collection Method | Data analysis type | MMAT score |
|---|---|---|---|---|---|---|
| Enane 2021 Kenya | To examine the burden of trauma among disengaged ALHIV in western Kenya and to investigate its potential role in HIV care disengagement | 42 HIV infected adolescents aged 10–19 years dis- engaged in care | Purposive sampling | Semi-structured interview guides | Through close reading and re-reading of the data and triangulation. | 80 |
| Jimu 2021 Zimbabwe | To contribute more robust strategies that can urgently inform innovative and targeted interventions that support ALHIV. | 30 HIV infected adolescents aged between 13 and 19 years. | Purposive sampling | 30 in-depth interviews and Stage 2: 15 semi-structured interviews and Stage 3: 5 interviews | An inductive approach was used to define themes, Thematic analysis | 100 |
| Kihumuro 2021 Uganda | To explore the experiences and factors surrounding anti-retroviral therapy adherence among adolescents in boarding schools. | 19 HIV infected adolescents aged 12–19 years | Purposive sampling of schools and purposive selection of 7 key informants per school | In-depth interviews each 30 to 40 minutes | Coded data analyzed thematically. | 100 |
| Kunapareddy 2014 Kenya | To identify key factors identified by HIV-infected adolescents on ART as contributing to medication adherence | 23 HIV infected adolescents aged 10–16 years. Eleven were female and 12 were male. | Convenience sampling | Three focus groups and three individual interviews were conducted using the same script of semi-structured interview questions | Coded in an initial stage of constant comparative analyses, Axial coding, sub-categories and organize themes into 'causal' relationships | 60 |
| MacCarthy 2018 Uganda | To assess barriers to ART adherence among HIV infected individuals aged 14–24 years to understand the unique challenges faced by this age group. | 11 HIV infected adolescents less than 18 years. | Purposive sampling | Focus group discussions | Content analysis | 80 |
| Madiba 2019 Botswana | To assess self-reported medication adherence among ALPHIV and explored structural factors that hinder or motivate them to adhere | 30 HIV infected adolescents aged 12–19 years | Purposive sampling | In-depth interviews | Thematic analysis was done following the approach of Braun and Clarke using both a deductive and an inductive approach to identify and report patterns or themes within the data. | 100 |
| Mutwa 2013 Rwanda | To better understand cART adherence barriers and successes in adolescents in Rwanda | 42 HIV infected adolescents aged 13–19 years | Not stated | Role-playing sessions, focus group discussions (FGDs) and in-depth interviews (IDIs) | The interviews were digitally recorded in Kinyarwanda, transcribed, and translated from Kinyarwanda into English, and uploaded into ATLAS.ti for analysis. | 80 |
| Orth 2021 South Africa | To explore experiences and challenges of being on ART, and individual interviews with 5 health workers to describe the challenges in treating ALHIV. | 18 HIV infected adolescents aged 10–17 years | Purposive sampling | Photovoice sessions; 6 Focus group discussions for adolescents and 5 individual interviews for HCWs | All interviews were audio-recorded, transcribed verbatim and translated where necessary. Photos were inserted into the transcripts, where these were referred to during photovoice discussion. All transcripts with photos were uploaded on Atlas.ti and subjected to thematic analysis. | 60 |

*(Continued)*

**Table 1.** (Continued)

| Study ID and settings | Aim/s of the study | Participants/ Population | Recruiting from (Sampling strategy) | Data collection Method | Data analysis type | MMAT score |
|---|---|---|---|---|---|---|
| Rencken 2021 South Africa | To explore the role of peer support in facilitating ART adherence and adoption of strategies to live successfully with HIV among South African ALHIV. | 35 HIV infected adolescents aged 10–19 years | Purposive sampling | Semi-structured interview | Structured thematic analysis using an iterative process. Transcripts were analyzed in NVivo12 software (QSR International Pty Ltd. Version 12, 2018 | 80 |
| Ritchwood 2020 South Africa | To identify aspects of the clinic environment that either improve or inhibit ALWH's ability to engage in HIV care. | 20 HIV infected adolescents aged 13 to 19 years | Purposive sampling | Semi-structured interviews | Inductive and deductive approaches to thematic analysis to identify, analyze, and report themes | 40 |
| Burns 2020 Malawi | To explore how social narratives and interactions within the everyday lives of ALHIV shaped their adherence behaviors | 45 participants; 16 HIV infected adolescents aged between 10 and 19 years, 13 caregivers, six health workers, and seven community members | Purposive sampling | In-depth interviews (IDI) were held with sixteen ALHIV, and four group activities incorporating participatory learning and action (PLA) tools | An inductive approach was used to define themes, inductively and iteratively using Nvivo11. | 100 |
| Appiah 2019 Ghana | To explores children's HIV disclosure experiences across Northern and Southern Ghana | 30 HIV infected adolescents aged 11 to 18 years | Purposive sampling | In-depth interviews | Phenomenological analysis which involves reading through data; organizing and coding; searching for patterns and interconnections; mapping and building themes; building thematic data; and, drawing conclusions | 100 |
| Apondi 2021 Kenya | To explore school-related barriers to adherence among students living with HIV (SLHIV) aged 13–17 years who had fully disclosed their HIV status in western Kenya. | 65 HIV infected adolescents aged 13 to 17 years | Purposive sampling | In-depth interview | Transcribed audio recordings verbatim. NVivo 8™ (QSR International) was used for coding and analysis. Thematic analysis was used, with data coded under one or more themes. | 100 |
| Denison 2015 Zambia | To explore ART adherence from the perspectives and experiences of older ALHIV (aged 15 18) and their adult caregivers in Zambia. | 32 HIV infected adolescents aged 15 to 18 years | Purposive sampling | 2 in-depth interviews per adolescent | Transcribed verbatim, translated, when necessary, into English, reviewed for accuracy and entered NVivo v.8 (QSR International). An iterative process was used to analyze the data with an initial codebook developed based on the interview guides and preliminary data and revised as the analysis team identified new themes | 100 |
| Ankrah 2016 Ghana | To identify barriers and facilitators to antiretroviral treatment adherence among adolescents in Ghana | 19 HIV infected adolescents aged 12 to 19 years | Purposive sampling | Semi-structured interviews | Interviews conducted in English were transcribed verbatim. Interviews in the local language were translated and transcribed into English with back-translation checks. Two researchers coded thematic analysis and data | 100 |

(*Continued*)

**Table 1.** (Continued)

| Study ID and settings | Aim/s of the study | Participants/ Population | Recruiting from (Sampling strategy) | Data collection Method | Data analysis type | MMAT score |
|---|---|---|---|---|---|---|
| vanWyk 2020 South Africa | To describe challenges to living with HIV and adherence to ART amongst school-going adolescents who receive ART at a public primary health care clinic in 2015–2016 in a low economic urban setting in the Western Cape province of South Africa | 14 HIV infected adolescents aged 10 to 19 years | Purposive sampling | Four focus group discussions (FGDs) and eight individual interviews | Content analysis; reading transcripts to coding and developing themes | 100 |
| Chory 2021 Kenya | To evaluate the feasibility and acceptability of a mobile-based mental health and peer support intervention using the WhatsApp® platform with ALWH in western Kenya | 15 HIV infected adolescents aged 10 to 19 years | Purposive sampling | In-depth interviews | Thematic analysis of the interviews, Preliminary codes were further refined through review of the coding structure | 100 |
| Cluver 2015 South Africa | Examines adherence-relevant experiences amongst adolescents whose HIV-positive status has been disclosed to them, as well as healthcare workers and caregivers in South Africa | 72 adolescents and 22 caregivers of HIV infected adolescents aged 10 to 19 years | Purposive sampling | 72 in-depth interviews (43 with adolescents: 26 interviewed once, 12 twice, 6 three times and 1 four times) 30 with 22 care givers | Qualitative interviews were recorded and transcribed in full or (where participants preferred not to be recorded) written notes were taken. Thematic codes were developed from the data, based on the principles of grounded theory, and were triangulated through cross-checking with both study participants and researchers on both methodological components of the study. | 60 |
| Falcão 2021 Mozambique | To explore the lived experiences of young ALHIV (12–14years) obtaining services in three health facilities in northern Mozambique | 14 HIV infected adolescents aged 12 to 14 years | Purposive sampling | In-depth interviews (IDI) | Deductive and inductive thematic analysis using QSR NVivo Version 12 | 100 |
| Mavhu 2013 Zimbabwe | To describe the internal and external life circumstances of HIV positive young people in Zimbabwe | 30 HIV infected adolescents aged 15 to 18 years | Purposive sampling | Three focus group discussions (FGDs) and 10 individual interviews | Tape-recorded qualitative data were transcribed and translated verbatim into English. Initial themes were identified during the data collection process; these were used to develop an initial coding framework. Additionally, sets of five IDIs, FGDs and RoLs were coded line by line on paper. Additional codes were added to the coding framework. Transcripts were then entered into NVivo 8 (QSR International, Melbourne, Australia), a qualitative data storage and retrieval program | 80 |
| Mesic 2019 Zambia | To identify, understand, and describe factors contributing to the losses in the latter stages of the CoC among ALHA in Zambia | 47 HIV infected adolescents aged 17 to 19 years | Purposive sampling | Six focus group discussions (FGDs) and 4 in-depth interviews | Qualitative analysis was conducted using QRS NVivo version 10.0 using thematic analytic techniques. | 100 |

**Table 1.** (Continued)

| Study ID and settings | Aim/s of the study | Participants/ Population | Recruiting from (Sampling strategy) | Data collection Method | Data analysis type | MMAT score |
|---|---|---|---|---|---|---|
| Nabukeera-Barunga 2015 Uganda | To describe the level and factors associated with adherence to antiretroviral therapy (ART) as well as the 1year retention in care among adolescents in 10 representative districts in Uganda | 227 HIV infected adolescents aged 10 to 19 years | Purposive sampling | 33 focus group discussions (FGDs) | Transcription of all the recordings was done. After that, all transcripts were translated into to English. Analysis of transcripts was done using Computer based analysis; Atlas-ti software | 100 |
| Nyogea 2015 Tanzania | To explore barriers and facilitators of ART adherence among children and teenagers in rural Tanzania | 35 HIV infected adolescents aged 13 to 17 years (mixed method study) | Purposive sampling | Two focus group discussions (FGDs) with adolescents and two FGDs with caregivers | Thematic content analysis was used to process all participants' descriptions along with identification of relevant concepts and ideas found in the transcripts linked to the topics of inquiry | 100 |
| Stangl 2021 Zambia | To assess the feasibility, acceptability and preliminary efficacy of a 6-session support group intervention designed to facilitate healthy transitions to adulthood among AGYW aged 15–19 living with HIV in Lusaka, Zambia. | 12 HIV infected adolescents aged 15 to 19 years | Purposive sampling | In-depth interviews | Interview transcripts and observation notes were coded in Nvivo 11.0 and analyzed using a content analysis approach. Deductive coding and analysis were employed | 100 |

## 2.4. Critical appraisal

Study quality was independently assessed by LDH and JLT through using methodological quality criteria outlined for qualitative, quantitative and mixed method studies in the Mixed Method Appraisal Tool (MMAT) [44]. Qualitative studies were assessed using the following five domains: a) appropriateness of the approach in answering research question b) adequacy of data collection methods to answer research question c) if findings were derived from data presented d) interpretation of results sufficiently validated by data and e) coherence between data sources, collection, analysis, and interpretation. Quantitative studies were assessed based on either as randomized control trials, non-randomized or descriptive studies. For randomized control trials the following domains were used; a) randomization, b) comparability of the groups, c) completeness of outcome data, d) blinding of assessors and e) adherence to intervention by participants. Non-randomized trials were assessed against a) appropriateness of the measurements, b) representativeness of target population, c) completeness of outcome data, d) controlling for confounder at design, analysis, and e) if intervention was administered as intended. On the other hand, descriptive quantitative studies were assessed for: a) relevancy of sampling strategy, b) representativeness of target population, c) appropriateness of measurements, d) level of risk of bias and e) appropriateness of statistical analysis. Lastly, mixed method studies were assessed for: a) rationale for used the design, b) integration of the two different components, c) interpretation of the outputs of the integrated components, d) if discrepancy and inconsistences between the two components were addressed and e) if the two components adhered to quality criteria of each component. The final scores for the assessment were agreed through discussions between LDH and JLT. PSN subsequently resolved conflicts via discussion with the dual screeners.

**Table 2. Characteristics of included quantitative studies.**

| Study ID and settings | Characteristics of study population | Study design | Interventions | ART adherence Barriers | ART adherence outcomes | MMAT score (%) |
|---|---|---|---|---|---|---|
| Abiodun 2021 Nigeria | 209 adolescents living with HIV. The median age (IQR): 16.00 (15.00–18.00) years. | A single-blind, parallel-design, and multi-center RCT | One SMS reminder each for follow-up appointments, 48 hours, and 24 hours before the follow-up visit date | Service provider retrieves leftover drugs after each visit forgetfulness, and social desirability | ART adherence rate Intervention group: 57/105 Control group: 47/104 Viral load (VL) suppression Intervention group: 63/105 Control group: 46/104 Unsuppressed VL Intervention group: 42/105 Control group: 58/104 | 100 |
| Bermudez 2016 Uganda | 761 HIV-infected adolescents from 10–16 years. | Cross-sectional | A family-based financial asset intervention on ARV adherence for youth living with HIV | Equity in ARV adherence by economic factors, and equity in ARV adherence by social factors. | ART adherence rate Good adherence: 349 Total population: 494 | 80 |
| Bermudez 2018 Uganda | 702 HIV infected adolescents aged from 10 to 16 years. | Cluster randomized trial | Family-based savings-led economic empowerment intervention | Economic insecurity and low resource environments. | Viral load (VL) suppression Intervention group: 218/331 Control group: 204/322 Unsuppressed VL Intervention group: 158/358 Control group: 130/344 | 60 |
| Bitwale 2021 Tanzania | 300 HIV infected adolescents age > 10 to 19 years | Cross-sectional study | None | Residence (urban versus rural), Caregiver marital status (married vs single), Disclosure to others, HIV status disclosed to child, Tuberculosis status | ART adherence rate Good adherence: 112 Total population: 183 Viral load (VL) suppression Suppression: 113 Total population: 198 Unsuppressed VL Unsuppressed: 48 Total population: 102 | 80 |
| Bongfen 2020 Cameroon | 455 HIV infected adolescents aged from 10–19 years | Cross-sectional study | None | Long waiting time and poor attitude of hospital staff, using traditional medicines was the socio-cultural factor, internalized stigma, being busy with other activities, having comorbid conditions, and not understanding the treatment regimen | ART adherence rate Good adherence: 279 Total population: 336 | 60 |
| Bulali 2018 Tanzania | 309 HIV infected adolescents aged from 10 to 17 years | Unmatched Case Control Study | None | HIV status disclosure status aged between 10–13 and 14–17, adolescent's level of education, Caregiver level of education | ART adherence rate Good adherence: 18 Total population: 227 | 100 |

*(Continued)*

**Table 2.** (Continued)

| Study ID and settings | Characteristics of study population | Study design | Interventions | ART adherence Barriers | ART adherence outcomes | MMAT score (%) |
|---|---|---|---|---|---|---|
| Bygrave 2012 Zimbabwe | 898 HIV infected adolescents and adults aged from 10–30 years. 306 Adolescents aged 10–19: Female 51% and 49% Males | Retrospective cohort study | None | Non-attendance to the clinic follow-up | Lost to follow up (LTFU) rate LTFU: 26 Total population:306 CD4 T cell count above 200 cells/μL CD4 above 200: 74 Total population: 306 | 80 |
| Haghighat 2021 South Africa | 1080HIV infected adolescents. 53,3% females and 44,7% males. | Longitudinal cohort | None | Rural residing | Lost to follow up (LTFU) rate LTFU: 183 Total population: 1080 Unsuppressed VL Unsuppressed: 513 Total population: 878 | 60 |
| Chaudhury 2018 Tanzania | 18,315 HIV infected adolescents aged from 10–19 years. Females: 59% and males: 41% | Retrospective cohort study | None | Delayed treatment initiation and loss of continuity of care. | Lost to follow up (LTFU) rate LTFU: 2515 Total population: 6908 | 100 |
| Crowley 2020 South Africa | 385 HIV infected adolescents aged 10–19. 58,2% Females and 41,8% Male | Cross-sectional | None | Missing a dose of ART because they fell asleep or were still sleeping. | ART adherence rate Good adherence: 76 Total population: 168 Viral load (VL) suppression Suppression: 226 Total population: 347 Unsuppressed VL Unsuppressed:121 Total population: 347 | 80 |
| Cluver 2021 South Africa | 969 HIV infected adolescents aged 10–19. 969: 55% Female and 45% Males | Cross-sectional | None | Perceived confidentiality at the clinic, travel to the clinic below 1 h, no emotional or physical violence victimization | ART adherence rate Good adherence: 405 Total population: 628 Lost to follow up (LTFU) rate LTFU: 63 Total population: 1046 | 100 |
| Cluver 2018 South Africa | 1059HIV infected adolescents aged 10–19 years. 55% Females and 45% males | Cross sectional | None | Age, gender, urban/rural location, formal/informal housing, maternal orphanhood, paternal orphanhood, mode of infection, time on ART treatment, and travel time to clinic. | ART adherence rate Good adherence: 764 Total population: 1059 Viral load (VL) suppression Suppression: 238 Total population: 1059 | 100 |

(*Continued*)

**Table 2.** (Continued)

| Study ID and settings | Characteristics of study population | Study design | Interventions | ART adherence Barriers | ART adherence outcomes | MMAT score (%) |
|---|---|---|---|---|---|---|
| Moyo 2020 Zimbabwe | 295 children and adolescents: 99 Adolescents | Cohort study | Enhanced Adherence Counselling | Being an adolescent | Viral load (VL) suppression Suppression: 21 Total population: 45 Non-suppression: 3 Total population: 14 | 60 |
| Wakooko 2020 Uganda | 1101 people living with HIV; 41 adolescents aged 13–19 years | Retrospective cohort | None | Anticipating and fear of side effects | Viral load (VL) un-suppression: Non-suppression: 13 Total population: 41 | 100 |
| Natukunda 2017 South Africa | 501adolescents aged 10–19; 54% Female and 46% Males; median age (IQR): 12–16. 14 | Cross-sectional | Combination ART | LPV/r-containing regimens, D4T-containing regimens, FDC of TDF + FTC + EFV was associated with reduced non-adherence. Past-month poor health, Food insecurity, Any opportunistic infections, clinic staff problems, and clinic transport problems | ART adherence: Good: 352 Total population: 501 | 80 |
| Umar 2019 Malawi | 209 Youth (13–24); 111 adolescents 13–16 | Cross-sectional | None | Every unit increase in HIV-related stigma, the odds of being virally unsuppressed increased by 1.08 (aOR 1.08, 95% CI 1.03–1.1.13). | Viral load (VL) suppression Suppression: 70 Total population: 111 Non-suppressed: 40 Total population:111 | 80 |
| Willis 2019 Zimbabwe | 100, 50 in each arm; Females- 60% in intervention and 62% in control; 40% males in intervention and 38% in control | RCT | Community adherence supporters (CATS) Vs Standard care | Forgetting to take their medications, pill fatigue, lack of adherence support, or they may be concealing their medication due to fears of stigma and discrimination. | ART adherence Intervention group: 34/47 Control group: 11/28 | 40 |
| Nasuuna 2018 Uganda | 449 children and adolescents; 192 adolescents | Retrospective cohort | Intense adherence counselling | Short time of counselling session at the health facility, the person delivering the message might also affect the message particularly the level of education and working experience. At patient level, there are multiple psychosocial factors that affect adherence which if not addressed could lead to the observed ineffectiveness of counselling. | Viral load Suppression Suppressed:34 Total population:116 Unsuppressed:82 Total population:116 | 80 |
| Jobanputra 2015 Eswatini | 12063 total people living HIV; 588 adolescents aged 10–19 years | Cross-sectional | Enhanced adherence counselling | Patients younger than 20, those with VL >1000 at initial test, and those with CD4 count <350 cells/ml were more likely to show virologic failure at retesting | Viral load: Unsuppressed: 33/98 | 60 |
| Gross 2015 Zimbabwe | 262; 61% Females and 39% males | Cross-sectional | Counselling | Attendance at group sessions run by a professional were protective against non-adherence, but group activities run by peers and individual counseling sessions were not associated with better adherence. | ART Adherence: Good adherence: 101/262 | 60 |

(*Continued*)

**Table 2.** (Continued)

| Study ID and settings | Characteristics of study population | Study design | Interventions | ART adherence Barriers | ART adherence outcomes | MMAT score (%) |
|---|---|---|---|---|---|---|
| Sithole 2018 Zimbabwe | 102 from each arm; Cases: Female- 53%, Male 47%; Controls: Female 62% and 38% Male | Unmatched case-control study | None | Poor adherence to ART was associated to taking alcohol and non- disclosure were the independent risk factors. | ART adherence: Cases: 26/102 Control: 31/102 Viral load suppression: Cases: 77/102 Control: 31/102 | 100 |
| Tanyi 2021 Kenya | 264 children and adolescents; 166 adolescents aged 10–19 | Retrospective study | None | While the overall EAC based on 1st viral load results were significant, Wald $\chi 2$ (2) = 37.173, p < 0.001, not doing EAC for those not virally suppressed based on the 1st viral load resulted in significant decreases of the log-odds of viral suppression, -6.683, p < 0.001 (95% CI: 0.000–0.011). | ART adherence: Good adherence: 95/ 125 Loss to follow up: 20/166 Viral load suppression: Suppressed: 76/109 | 60 |
| Desta 2020 Ethiopia | 19532 people living with HIV; 420 adolescents aged 15–19 years | Retrospective cross- sectional study | None | Patients whose viral load was determined for suspected ART failure initial VL compared to routine first VL is expected to have a rate (IRR = 0.69, 95% CI = 0.58– 0.81) times lower for the recent CD-4 T-cells count. Patients with a regimen 1e (TDF-3TC-EFV) compared to a 1c (AZT-3TC-NVP) regimen. | Viral load suppression: Suppressed; 14372/ 19525 CD4 T cell CD4 count<200: 3043/19525 | 60 |
| Firdu 2017 Ethiopia | 273: 52.7% Female and 47,3% Male adolescents aged 13–19 years | Cross-sectional | None | WHO stage IV) and adolescents with widowed parent to have an independent and statistically significant association with optimum ART adherence | ART adherence: Good adherence: 216/273 | 80 |
| Kabogo 2018 Kenya | 319; Male 51,4% and 48,6% Female children and adolescents aged 0–18 | Retrospective cohort | Universal Test and Treat Vs Pre UTT | Universal Test and Treat (UTT) period, both guardian caregiver status and being 11–14 years old were predictive of suboptimal adherence: UTT guardian homes: univariate HR = 2.27, 95% CI 1.12–6.72, P = 0.041); children aged 11–14 years: Univariate HR = 2.16, 95% CI 1.03–5.73, P = 0.018. | ART adherence: Good adherence: 129/319 CD4 Count CD4>200: 261/319 Viral load suppression: Suppression: 232/ 319 | 80 |
| Bvochora 2019 Zimbabwe | 656; 55% Female and 45% Male; 85 adolescents aged 10–19 | Retrospective cohort | Enhanced Adherence Counselling | CD4 cell count >350 cells/mm3; being on 2nd line ART regimen at the time of initial high viral load test result, initial viral load levels >5000 copies per ml. | Viral load suppression: Un-suppressed: 18/ 53 | 40 |
| Zhou 2021 South Africa | 933; Females 55% and Male 45% adolescents aged 10–19 | Retrospective cohort | None | Inconsistent adherence was significantly associated with older age and horizontal HIV acquisition. | ART adherence: Good adherence: 346 Total population:933 | 60 |
| Denison 2020 Zambia | 279: 139 Intervention and 137 Comparison; 174 adolescents aged 15 to 19; 87 in intervention and 87 in control arm | RCT | Project Yes Vs Standard Care | Odds of having these feelings (guilt, shame and worthless) were significantly reduced by a factor of 0.39 relative to the reduction in the comparison arm [interaction terms OR:0.39, 95% CI: 0.21,0.7]. | Viral load suppression: Suppressed: Intervention group: 31/44 Control group: 38/ 64 | 40 |

(*Continued*)

**Table 2.** (Continued)

| Study ID and settings | Characteristics of study population | Study design | Interventions | ART adherence Barriers | ART adherence outcomes | MMAT score (%) |
|---|---|---|---|---|---|---|
| Munyayi 2020 Namibia | 385: 53.2% Male and 46,8 Female; 78 Intervention and 307 Standard care adolescents aged 10–19 | Retrospective cohort | Teen Clubs Vs Standard Care | Being an older adolescent | ART adherence: Good adherence/ Exposed group: 74/ 87 Non-exposed group:276/296 Viral Load suppression Suppressed: Exposed group: 44/ 70 Non-exposed: 175/ 259 Unsuppressed: Exposed group: 14/ 70 Non-exposed group: 28/259 | 60 |
| Natukunda 2019 Uganda | 238; 52% Male and 48% Female adolescents aged 10–19 | Cross-sectional | None | History of treatment failure and had normal nutritional status, being Anglican or Muslim religions. | ART adherence: Good adherence 160/200 Viral load suppression: Suppressed: 131/200 | 60 |
| Ndiaye 2013 Botswana | 82; 35% Males and 65% Females adolescents aged 13 to 18 years | Cross-sectional | None | Male sex (OR 3.29, 95% CI 1.13–9.54; P = 0.03) was the only factor which was independently and significantly associated with suboptimal ART adherence. | ART adherence: Good adherence: 62/82 | 60 |
| Chawana 2017 Zimbabwe | 50: 54% Females and 46% Males adolescents aged 10 to 18 | RCT | Modified directly administered ART and standard care Vs Standard care and self-administered treatment for 90 days | Common reasons for missing ART were simply forgetting (68%), being away from home (62%), problem with keeping time (50%) and falling asleep before taking medication or waking up late (46%). | ART adherence: Good adherence: Intervention group: 15/23 Control group: 10/ 27 Viral load suppression: Suppressed: Intervention group: 12/23 Control group: 8/27 | 60 |
| Mburu 2019 Kenya | 2195; 58,1 Females and 41,9% Males aged 10–19 | Retrospective cohort | APOC (Adolescent's package of care) training | At clinics offering adolescent-friendly clinic days, adolescents were 2 times more likely to be virally suppressed than at facilities not offering these specialized clinic days. | Viral load suppression: Post intervention: 958/1345 Pre-intervention: 1345/2195 | 60 |
| Matyanga 2016 Zimbabwe | 1594 patient; 110 (Female 43.6% Male and 56,4%) adolescents aged 10–19 | Retrospective cohort | Integrated HIV and TB services | Among adolescents, initiation on ART with WHO Stage IV (Compared to Stage I) was associated with more attrition | Loss to follow up: 28/110 CD4 T cell Count CD4<200: 32/110 | 80 |
| Van Wyk 2020 South Africa | 220: 82,7% Females and 17,3% Males adolescents aged 10–19 | Retrospective cohort | None | Older adolescents (15–19 years) were observed to have lower rates of VLS than younger adolescents and being female adolescent. | Loss to follow up: 80/220 Viral load suppression: Suppressed: 55/220 CD4 T cell count CD4<200: 14/55 | 80 |

(*Continued*)

**Table 2.** (Continued)

| Study ID and settings | Characteristics of study population | Study design | Interventions | ART adherence Barriers | ART adherence outcomes | MMAT score (%) |
|---|---|---|---|---|---|---|
| Gitahi-Kamau 2020 Kenya | 82: 61% Male and 39% Females adolescents aged 16–19 | Cross-sectional | None | Older ALWHIV with high ART adherence self-efficacy were eight times more likely to report high adherence and those on a twice a day ART regimen were almost four times more likely to report adherence. Self–esteem, age, sex, schooling, perceived social support and experienced stigma were not found to be associated with adherence among these older adolescents. | ART adherence Good adherence: 52/81 CD4 T cell count CD4<400: 53/81 | 20 |
| Vogt 2017 Zimbabwe | 1499; 48% Males and 52% Female adolescents aged 10–19 | Retrospective cohort | None | High mortality and LTFU shortly after becoming eligible, especially among adolescents not on ART, | Loss to follow up: 133/1499 | 60 |
| Jerene 2019 Ethiopia | 2058: 53,4%Females and 46,6%Males adolescents aged 10–19 | Retrospective cohort | None | Being female, having a rural residence and lower CD4 count at baseline predicted higher LTFU rate | Loss to follow up: 209/1531 | 100 |
| Meloni 2020 Nigeria | 476; 50,8% Females and 49,2 Males adolescents aged 10–19 | Retrospective cohort | None | Both adherence counseling and monitoring for adolescents may require different methods than what is typically used for adults. Adherence counseling for adolescents should be effective and not punitive such that some adolescents may feel a need to hide poor adherence. | ART adherence Good adherence: 405/476 Loss to follow up: 57/476 Viral load suppression Suppressed: 121/189 | 40 |
| Okonji 2021 South Africa | 9386; 55% Females and 45% Males adolescents aged 10–19 | Cross-sectional | None | ALHIV on second line treatment were less likely to attain viral suppression compared to their reference group (aOR = 0.41, 95% CI 0.34–0.49). | Viral load suppression Suppressed: 6975/9386 Un-suppressed: 2411/9386 CD4 T Cell count CD4<200: 1071/9386 | 60 |
| Brathwaite 2021 Uganda | 656 adolescents aged 10–16 | Longitudinal study | None | Family cohesion, Child poverty, Economic intervention | ART adherence Good adherence: 404/637 | 60 |
| Chory 2021 Kenya | 29 HIV infected adolescents aged 10 to 19 | Cohort | Mobile-based mental health and peer support intervention using the WhatsApp platform | Perceived stigma, depression, and anxiety | ART adherence rate: 17/29 | 80 |
| Cluver 2015 South Africa | 684 HIV infected adolescents aged 10 to 19 | Cross-sectional | None | Full adherence was negatively associated with being older (OR 0.85; 95% CI 0.80–0.91, P<0.001) and longer travel time to the clinic (OR 0.72; 95% CI 0.57–0.89, P<0.005) | ART adherence rate: 438/684 | 100 |
| Dulli 2018 Nigeria | 30 HIV infected adolescents aged 15 to 19 | Cohort | Structured support group intervention—SMART (social media to promote Adherence and Retention in Treatment) Connections which is delivered through a social media | Stigma and low engagement level during scheduled intervention | Viral load (VL) suppression: 15/30 Unsuppressed VL: 15/30 | 60 |

(*Continued*)

**Table 2.** (Continued)

| Study ID and settings | Characteristics of study population | Study design | Interventions | ART adherence Barriers | ART adherence outcomes | MMAT score (%) |
|---|---|---|---|---|---|---|
| Falcão 2021 Mozambique | 61 HIV infected Adolescents aged 12 to 14 years 61; 49% males and 51% Females | Cross-sectional | None | Forgetfulness was the biggest challenge. 3% reported growing tired of taking pills was another challenge. | Viral load (VL) suppression: 17/37 | 100 |
| Mesic 2019 Zambia | 330 HIV infected adolescents aged 18 to 19 years. 48% males and 52% Females | Cross-sectional | None | Missing an ART clinic appointment in the past 3 months because they had to walk site of appointment and employment. ALHA reported that forgetting the appointment, lacking transport (, and family commitments were the most common reasons they missed appointments. | ART adherence rate: 180/220 | 80 |
| Nabukeera-Barungi 2015 Uganda | 1842 HIV infected adolescents aged 10–19: 63% Females and 37% Males | Cross-sectional | None | Rural health facilities being associated with poor adherence. | ART adherence rate: 1588/1842 | 100 |
| Stangl 2021 Zambia | 21 HIV infected adolescents aged 15–19 | Cross-sectional | Support group intervention | Last of ARV knowledge | ART adherence rate: 13/14 | 60 |

## 2.5. Statistical analysis

Data analysis was done through parallel results convergent design where qualitative and quantitative findings were analyzed separately and integrated in the overall results. For *quantitative analysis*, a meta-analysis was conducted using Stata software version 16 and Open Meta analyst. Meta-analysis was performed to calculate the pooled estimates of the following outcomes: ART adherence, viral load suppression, unsuppressed viral load, and loss to follow up. The pooled effect size was calculated as rates and the odds ratios and their 95%CIs. As we expected high heterogeneity between studies, subgroup analysis was undertaken based on different ART barriers reported in the studies. In addition, we also performed meta-regression based on the adolescents' age groups. A random effect meta-analysis was used to pool the estimates because of the variability of study designs and interventions. Heterogeneity was quantified using the $I^2$ to identify study variation. Furthermore, publication bias was used to assess to the funnel plot symmetry and the Egger rank and Begg correlation tests were used to assess the quantitative publication bias. For statistical analysis, P-values of 0.05 were considered statistically significant. As Stata software version 16 reported the Log OR, we converted this to the OR.

*Qualitative analysis* was conducted through grouping key finding into the main themes according to both the barriers and facilitators to ART adherence as relevant to the study (Tables 4 and 5). Similarly, barriers to ART adherence were analyzed for *quantitative* design studies narratively through grouping findings into main barriers to ART themes (Table 3). The themes used in this study were patient-related, social, therapy-related, health system-related, economic, and cultural barriers [9]. These themes were then subjected to a meta-synthesis to produce a single comprehensive set of synthesized findings.

## 3. Results

A total of 10 431 studies were found after running the search strategy in the different databases and 4744 duplicates were removed. The process followed the PRISMA guidelines as reported

**Table 3. Barriers to anti-retroviral therapy adherence among adolescents aged 10 to 19 years living with HIV in sub-Saharan Africa (Quantitative studies).**

| Barriers | Key words | Studies |
|---|---|---|
| Economic barriers | Poverty, economic insecurity, low resource environments, Food insecurity | [Bermudez 2016; Bermudez 2018; Natukunda 2017; Natukunda 2019; Brathwaite 2021] |
| Social barriers | social desirability, equity in ARV adherence by social factor, location (urban versus rural), Caregiver marital status, Status disclosure to others or to self, internalized stigma, Level of education, care-giver level of education; Emotional or physical violence victimization, Age, gender, orphanhood, mode of infection, stigma, guilt, shame, worthlessness, sex (male); perceived support, self-esteem, family cohesion, lack of transport, living with non-parent care-taker, having a single parent, being a member of a social support group | [Abiodun 2021; Bermudez 2016; Bitwale 2021; Bongfen 2020; Bulali 2018; Haghighat2021; Cluver 2021; Cluver 2018; Moyo 2020; Umar 2019; Willis 2019; Nasuuna 2018; Jobanputra 2015; Firdu 2017; Kabogo 2018; Bvochora 2019; Zhou 2021; Denison 2020; Munyayi 2020; Ndiaye 2013; van Wyk 2020; Gitahi-Kamau 2020; Jerene 2019; Brathwaite 2021; Falcão 2021; Cluver 2015; Mesic 2019;] |
| Health system barriers | Long waiting time, attitude of healthcare workers, Not understanding treatment regime or lack of ART knowledge, delayed treatment initiation, Perceived confidentiality at clinic, travelling long distance to clinic; time taken to conduct counselling session, working experience and knowledge of health worker, counselling session (group or individual) run by health professional or peer, Universal Test and treat period, type of clinic (adolescent friendly), lack of follow up, rural health facility | [Bongfen 2020; Chaudhury 2018; Cluver 2021; Cluver 2018; Stangl 2021; Natukunda 2017; Nasuuna 2018; Gross 2015; Sithole 2018; Kabogo 2018; Bvochora 2019; Mburu 2019; Vogt 2017; Meloni 2020; Cluver 2015; Mesic 2019; Nabukeera-Burungi 2015; Stangl 2021] |
| Therapy-related barriers | time on ART, side effects, ART regimen, pill fatigue, CD4 Count at initiation, first viral load test result, WHO HIV stage, viral load status; | [Cluver 2018; Wakooko2020; Willis 2019; Jobanputra 2015; Tanyi 2021; Desta 2020; Firdu 2017; Bvochora 2019; Natukunda 2019; Matyanga 2016; Gitahi-Kamau 2020; Jerene 2019; Okonji 2021; Falcão 2021] |
| Patient-related barriers | Forgetfulness, being busy, non-attendance of clinic follow-up appointments, loss of continuity of care, being sleepy, other co-morbidities, Poor health, alcohol use, being away from home, waking up late, Time on ART; CD4 Count at initiation, first viral load test result; WHO HIV stage, viral load status | [Abiodun 2021; Bongfen 2020; Bygrave 2012; Chaudhury 2018; Crowley 2020; Natukunda 2017; Willis 2019; Sithole 2018; Chawana 2017; Falcão 2021; Cluver 2018; Jobanputra 2015; Tanyi 2021; Firdu 2017; Natukunda 2019; Matyanga 2016; Jerene 2019; Okonji 2021]] |
| Cultural barriers | Use of traditional medicine, religion, being prayed for, | [Bongfen 2020; Natukunda 2019] |

in Fig 1. Only 206 full-text studies met screening criteria and 66 studies retained and were included in the analysis: quantitative studies (n = 41), qualitative studies (n = 16), and mixed study (n = 9). Other studies were excluded because of wrong data collection period, wrong population group, non-disintegrated by age and one study was excluded because of its small sample size.

## 3.1. Characteristic of included studies

Out of the 66 studies included, 24 studies were included for the qualitative analysis of which 8 (out of the 9 mixed methods studies, 8 were included in the qualitative analysis as one study did not report on the outcome of interest in the qualitative component but only the quantitative component) of these were the qualitative component of mixed-method studies as reported in Table 1. Most studies (7/24) included adolescents aged 10 to 19 years followed by 13 to 19-year-old (4/24) and 12 to 19-year-old (3/24). Majority (5/24) of the studies were conducted in South Africa followed by Kenya (4/24). The other studies originated from the following

**Table 4. Barriers to ART adherence (Qualitative studies).**

| Theme | Sub-theme | Codes | Study ID |
|---|---|---|---|
| 1. Social | a. Stigma and discrimination | i) Lack of privacy<br>ii) Unintended disclosure<br>iii) Isolation, depression, suicidality<br>iv) Mental health disorder | [Enane 2021; Jimu 2021; Kihumuro 2021; Kunapareddy 2014; MaCarthy 2018; Madiba 2019; Mutwa 2013; Orth 2021; Appiah 2019; Apondi 2021; Denison 2015; Ankrah 2016; van Wyk 2019; Chory 2021; Mavhu 2013; Mesic 2019; Nabukeera-Barunga 2015; Stangl 2021; Nyogea 2015] |
| | b. Social support | i) School mechanism (scheduling of classes, privacy)<br>ii) Caregiver | [Kihumuro 2021; Kunapareddy 2014; MaCarthy 2018; Madiba 2019; Ritchwood 2020; Burns 2020; Appiah 2019; Apondi 2021; vanWyk 2020; Cluver 2015; Mavhu 2013; Mesic 2019; Nabukeera-Barunga 2015; Nyogea 2015] |
| 2. Economic | a. Poverty | i) Access to water and food<br>ii) Transport money | [Kihumuro 2021; Madiba 2019; Appiah 2019; Ankrah 2016; van Wyk 2019; Mesic 2019; Nabukeera-Barunga 2015; Mavhu 2013] |
| 3. Health system related | a. Clinic factors | i) Distance to clinic<br>ii) Waiting time<br>iii) Overcrowding<br>iv) Not able to refill (stock-out)<br>v) Inflexibility of appointment scheduling | [Madiba 2019; Mutwa 2013; Ritchwood 2020; van Wyk 2019; Nyogea 2015; Mavhu 2013] |
| | b. Health workers factors | i) Confidentiality<br>ii) Singling out/ lack of sensitivity<br>iii) Missing or misplaced files<br>iv) Attitude of health care workers | [Ritchwood 2020; Burns 202; van Wyk 2019; Nyogea 2015] |
| 4. Therapy related | a. Regime | i) Frequency of taking pills | [Mutwa 2013] |
| | b. Side effect | i) Difficult to swallow<br>ii) Dizziness<br>iii) nausea | [Madiba 2019; Orth 2021; Ankrah 2016; Nabukeera-Barunga 2015] |
| | c. Other | iv) Container | [Kihumuro 2021; Apondi 2021] |
| 5. Patient related | a. Attitude | i) Rebellious attitude towards HIV medication | [Jimu 2021; Burns 2020; Falcão 2021; Mesic 2019] |
| | b. Responsibility for medication | i) Forgetting<br>ii) Oversleeping<br>iii) Not being home<br>iv) Busy<br>v) Taking medication late<br>vi) Treatment fatigue<br>vii) Being drunk | [Kihumuro 2021; Mutwa 2013; Kunapareddy 2014; MaCarty 2018; Apondi 2021; Denison 2015; Ankrah2016; vanWyk 2020; Falcão 2021; Mesic 2019; Nyogea 2015] |
| | c. Knowledge | i) Myths and misconceptions<br>viii) HIV status | [Denison 2015; Nabukeera-Barunga 2015] |
| 6. Cultural | Religion and culture | i) Spirituality and healing<br>ii) Traditional medicine | [Denison 2015; Nyogea 2015] |

countries: Zambia (3), Uganda (3), Zimbabwe (2), Ghana (2), Rwanda (1), Malawi (1), Botswana (1), Tanzania (1), Mozambique (1). Of all the 24 studies, 10 used in-depth interviews to collect data while 7/24 used both focus group discussions and interviews for data collection. Semi-structured interviews were used by 4 studies while 3 used focused group discussions only.

Among the quantitative studies (Table 2), 48 studies were included in the meta-analysis of which 7 were the quantitative component of the mixed method studies (two mixed method studies were not included in the quantitative analysis since in their quantitative component, study population was not disaggregated by age but were only disaggregated in the qualitative component was included in this study). Majority of the studies (29) studies included adolescents aged 10 to 19 followed by 4 studies with adolescents aged 15 to 19 years. Twenty-one studies were cross-sectional in design while 19 were cohort studies. Nine studies were conducted in South Africa and another 9 in Zimbabwe followed by 7 studies conducted in Uganda. The other studies were conducted in the following countries: Kenya (5); Zambia (3);

**Table 5. Facilitators of ART adherence.**

| Theme | Sub-Theme | Code | Studies ID |
|---|---|---|---|
| 1. Support | a. Peer | Peer mentoring | [Enane 2021; MaCarthy 2018; Rencken 2021; Mesic 2019; Nabukeera-Barunga 2015; |
| | b. Parents/ family | i) Reminders | [Kihumuro 2021; Madiba 2019; Mutwa 2013; Orth 2021; Denison 2015; Ankrah 2016; van Wyk 2019; Falcão 2021; Mavhu 2013; Mesic 2019] |
| | c. School | School mechanism | [Kihumuro 2021; Mabiba 2019; Nabukeera-Barunga 2015] |
| | d. Health facility | i) Adolescent friendly practices<br>ii) Co-located community HIV clinics<br>iii) Adequate supply | [Jimu 2021; Kihumuro 2021; Orth 2021; Ritchwood 2020; Burns 2020; Denison 2015; Ankrah 2016; Mesic 2019; Nabukeera-Barunga 2015] |
| | e. Personal | i) Sense of hope<br>ii) Reminders | [Madiba 2019; Appiah 2019; Denison 2015; Ankrah 2016; Chory 2021; Mesic 2019; Nabukeera-Barunga 2015; Stangl 2021] |
| | f. Financial | iii) Transport money | [Jimu 2021; van Wyk 2019; Nabukeera-Barunga 2015] |
| 2. Secrecy or Confidentiality | a. Storage of medication | i) Storages/ lockers in boarding school<br>ii) Container of medication | [Kunapareddy 2014; Ankrah 2016; Kihumuro 2021] |
| | b. Status Disclosure | i) To HIV positive peers<br>ii) To sero-negative peers<br>iii) others | [Kihumuro 2021; Kunapareddy 2014; MaCarthy 2018; Mutwa 2013; Orth 2021; Burns 2020; Cluver 2015] |
| 3. Counselling and education | a. ART knowledge | Benefits of ART (Comprehensive sexuality and HIV and AIDS education | [Jimu 2021; Madiba 2019; Ritchwood 202; Ankrah 2016; Mesic 2019; Nyogea 2015; Stangl 2021] |

Ethiopia (3); Tanzania (3); Nigeria (3); Eswatini (1); Botswana (1); Namibia (1); Mozambique (1); Cameroon (1) and Malawi (1). The distribution of the studies including ART adherence among adolescent living with HIV and on ART is shown in Fig 2 below.

## 3.2. Quality assessment

The results of the quality assessment are shown in the S2 Table. For each domain, the score was "yes or no," and a percentage of the "yeses" was used as the final score for each study.

**Fig 2. Map showing the studies included in the review.**

Among the qualitative studies, only one study scored less than 50% [45], two scored 60% [44, 45] and the remaining thirteen were scored 80 and above [41, 46–57]. Among the quantitative studies, four were scored below 50% [58–61]. Seventeen scored 60% [62–78] and nineteen scored 80% and above [72, 79–96]. Lastly, among the mixed studies, two studies scored 60% [97, 98] and the remaining seven scored 80% and above [28, 99–104].

### 3.3. Interventions

A total of 13 interventions were identified in 19 of the quantitative studies included [58, 59, 65, 66, 70, 71, 74, 77, 79, 90, 93, 94, 97, 98, 105–109]. These were as follows: SMS reminders [81]; family based financial asset [64]; enhanced counselling [59, 65, 66, 70, 90]; combination ART [107]; community adherence supporters [60]; universal test and treat [95]; peer mentoring [108]; teen clubs [73]; modified directly administered ART [76]; adolescents package of care [77]; integrated HIV and TB services [96]; mobile-based mental health [109] and peer support intervention and support groups [97, 98].

### 3.4. Meta-analysis

**3.4.1. ART adherence.** ART adherence was reported in 22 studies from 11 countries in SSA (Fig 3). The overall ART adherence rate was 65% (95% CI 56–74; $I^2$ 99.01%) among adolescents on ART [62, 63, 65, 66, 69, 71, 72, 74, 75, 80, 82–84, 87, 89, 94, 95, 103, 104, 107, 109, 110]. The lowest ART adherence rate was reported in a study conducted in Tanzania while the highest rate was reported in a study conducted in Uganda (8% (95% CI 4–11)) and (93% (95% CI 79–100)) respectively [84, 104] (Fig 3 and S1 Fig).

Effects of ART adherence interventions showed that SMS reminders, community adherence supporters, teen clubs and modified directly administered ART were not statistically associated with ART adherence (OR 1.06 (95% CI 0.85–1.35, p = 0.62 $I^2$ 0.00%)) [60, 73, 76, 81, 93] (Fig 4). ART adherence was 1.2 (95% CI 0.75–1.93) higher among those reporting a SMS reminder compared to those not receiving a SMS reminder [81] (Fig 4). Among adolescents who received standard care and had access to a community adherence support intervention, ART adherence was 1.84 (95% CI 0.81–4.22) compared to adolescents only receiving standard care [60]. Modified directly administering of ART was associated with 1.77 (95% CI 0.66–4.66) ART adherence while teen clubs and a viral load >1000c/ml were negatively associated with ART adherence (OR 0.95 (95% CI 0.64–1.30)) and (OR 0.84 (95% CI 0.47–1.51)) respectively [73, 76, 93] (Fig 4).

**3.4.2. Barriers to ART adherence.** Six categories of barriers to ART were reported in included studies. These were social, economic, health systems-related, therapy-related, patient-related and cultural barriers as adopted from previous studies [9]. Table 3 shows the barriers presented in the included studies.

ART adherence was high among studies reporting therapy-related barriers (76% (95% CI 68.5–83.5)) followed by studies reporting health systems-related barriers (75.3% (95% CI 58.3–92.3; $I^2$ 98.7%)) [63, 69, 72, 100, 104] (S1 Fig). ART adherence was low among studies reporting social barriers (45.2% (95% CI 19.9–70.4; $I^2$ 99.1%)) and patient reported barriers (45.2% (95% CI 37.7–52.8)) [71, 75, 83, 84, 87]. However, ART adherence was reduced in studies reporting combination of health systems-related and patient-related barriers (64% (95% CI 60.4–67.6)), social and health systems-related barriers (52.5% (95% CI 28.9–76.1; $I^2$ 98.1%)); social and patient-related barriers (58.6% (95% CI 40.7–76.5)) and economic and social barriers (63.4% (95% CI 59.7–67.2)) [80, 88, 95, 109, 110] (S2 Fig). ART adherence was high among studies reporting a combination of social, health systems-related, patient-related and cultural barriers (83% (95% CI 79–87.1)); social, health systems-related and therapy-related barriers

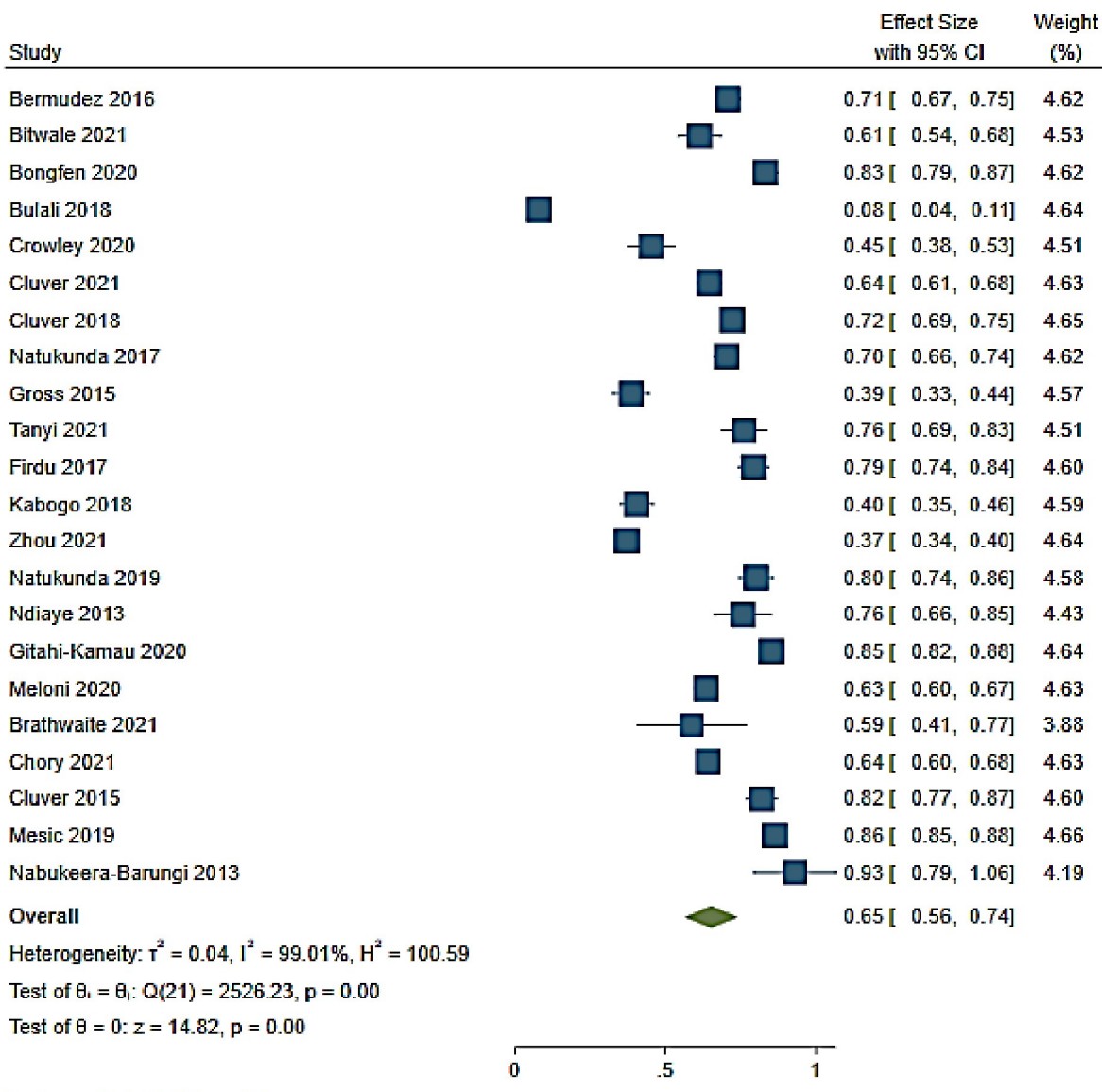

**Fig 3. Forest plot of ART adherence rate among adolescents aged from 10 to 19 years.**

(72% (95% CI 769.4–74.9));social and therapy-related barriers (72.4% (95% CI 57.9–87; $I^2$ 84.5%)) [62, 65, 89, 94]; economic, social and therapy-related barriers (81.8% (95% CI 74.5–85.5)); economic and social barriers (70.7% (95% CI 66.6–74.7)) and economic, health systems-related and patient-related barriers (70.3% (95% CI 66.3–74.3)) [82, 103, 107] (S1 Fig).

**3.4.3. Viral load suppression.** Fifteen studies were included in the analysis of the viral load suppression (VLS) rate among adolescent in SSA [63, 67, 69, 74, 79, 83, 87, 89, 91, 92, 95, 97, 99, 102, 111]. The overall viral load suppression rate was 55% (95% CI 46–64: $I^2$ 99.5%) (Fig 5). VLS was low among studies reporting social and therapy-related barriers (22% (95% CI 20–25; $I^2$ 98.7%)) and social barriers (48% (95% CI 31–61)) [67, 83, 89, 91, 97] (Fig 5). However, VLS was high among studies reporting therapy-related barriers (74% (95% CI 73–75; $I^2$ 22.2%)); economic, cultural and therapy related-barriers (65% (95% CI 59–72)); health system-related barriers (64% (95% CI 57–71)); patient-related barriers (56% (95% CI 43–64; $I^2$

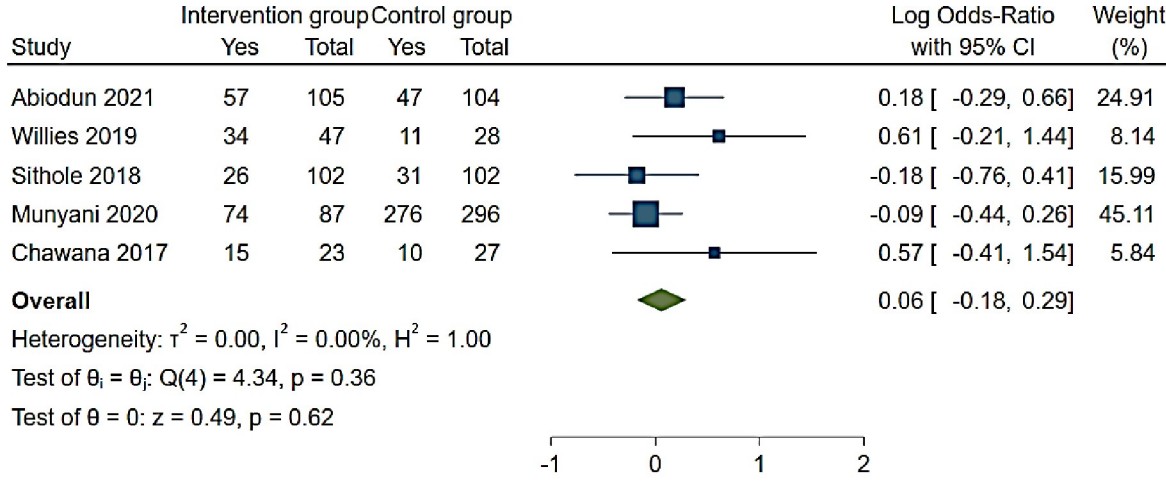

**Fig 4. Forest plot of ART adherence interventions among adolescents aged from 10 to 19 years.**

68.1%)) and social, health system-related and therapy-related barriers (50% (95% CI 9%-94%)) [63, 69, 70, 74, 79, 87, 92, 95, 99,102] (Fig 5).

Additionally, seven studies reported on interventions reporting ART adherence barriers and VLS [64, 73, 76, 77, 81, 93, 108]. VLS was not statistical associated with the barriers to ART (OR 1.25 (95% CI 1–1.55 p = 0.05: $I^2$ 61.1%)). However, in a study conducted in Zimbabwe, health systems and therapy-related barriers were associated with over two-fold increase in viral load suppression (OR 2.48 (95%CI 1.51–4.10)) [93]. Economic (OR 1.04 (95%CI 0.81–1.32)) [64], social (OR 1.01 (95%CI 0.71–1.42; $I^2$ 0.00%)) [73, 108], health systems-related (OR 1.16 (95%CI 1.041.30)) [93]; patient-related (OR 1.77 (95%CI 0.61–5.05)) [76] and social and patient-related barriers (OR 1.35 (95%CI 085–2.16)) [81] were not associated with VLS (Fig 6).

**3.4.4. Unsuppressed viral load.** A total of 11 studies assessed barriers to ART adherence and unsuppressed viral load rate among adolescents [61, 66–68, 79, 83, 87, 90–92, 99] (Fig 7). The overall unsuppressed viral load rate was 41% (95%CI 32–50; $I^2$ 96.5%) for included studies (Fig 7). The included studies reported the following barriers: social [66, 67, 83, 91]; therapy-related [79, 90]; patient-related [87, 99]; social and health system-related [92]; social and therapy-related [68] and lasty social, health system-related and therapy-related barriers [61] (Fig 7). Unsuppressed viral load rate was high in a single study reporting social and health system-related barriers (71% (95%CI 62–79)) [92] followed by studies reporting social barriers and patient-related barriers (43% (95% CI 29–57; $I^2$ 90.9%)); (40% (95% CI 26–54; $I^2$ 60.7%)) [87, 99] respectively [66, 67, 83, 87, 91, 99] (Fig 7). Unsuppressed viral load was noted in studies reporting therapy-related barriers (26% (95% CI23-27; $I^2$ 0.02%)) [79, 90] as well as in a study reporting social and therapy-related barriers (34% (95% CI 24–45)) [68] (Fig 7).

Moreover, three studies reported interventions for adolescents associated with unsuppressed viral load [64, 73, 81] (Fig 8). The odds of VLS among adolescents receiving intervention were significant reduced compared to controls (OR 0.44 (95% CI 0.36–0.54: $I^2$ 0.00%)) (Fig 8).

**3.4.5. Loss to follow up.** Among the 10 studies included, loss to follow up (LTFU) was reported to be 17% (95% CI 10–24; $I^2$ 99.1%) [63, 66, 69, 78, 85, 86, 96–98, 112] (Fig 9). LTFU was high (36% (95% CI 36–38)) among adolescents in a single study reporting health system and patient-related barriers [86] followed by studies reporting social barriers at 26% (95% CI 7–45; $I^2$ 96.9%) [66, 97] (Fig 9). Therapy-related barriers were associated with 18% (95% CI

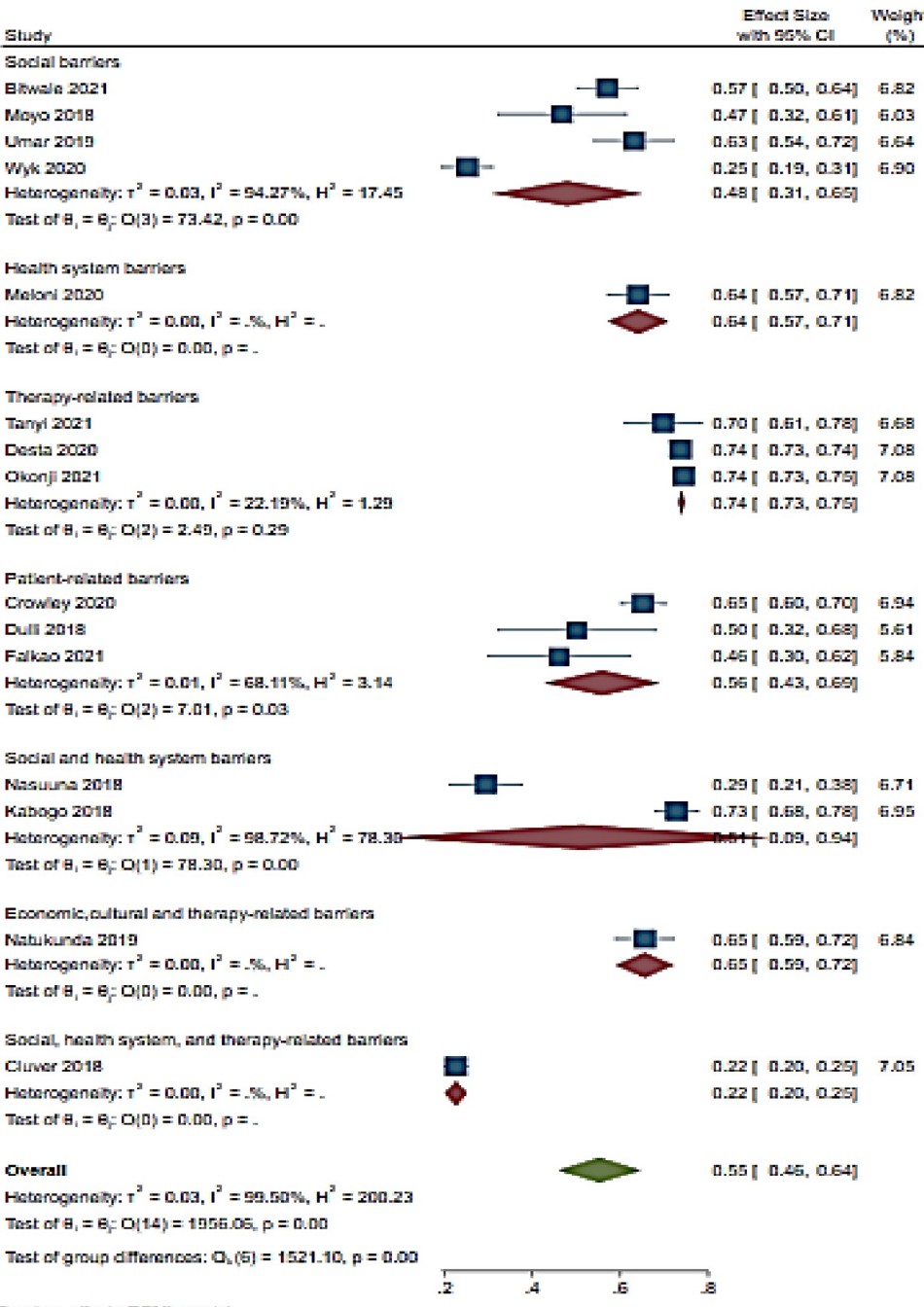

**Fig 5. Forest plot of viral load suppression and ART adherence barriers among adolescents aged from 10 to 19 years.**

5–31; $I^2$ 86.4%) [69, 96] LTFU while social and therapy-related was associated with 14% (95% CI 12–15) [98] (Fig 9). LTFU was less (6% (95% CI 5–9)) in a study conducted in South Africa reporting social and health system-related barriers followed by a study conducted in Zimbabwe reporting patient-related barriers with 9% (95% CI 5–12) LTFU rate [85, 112] (Fig 9).

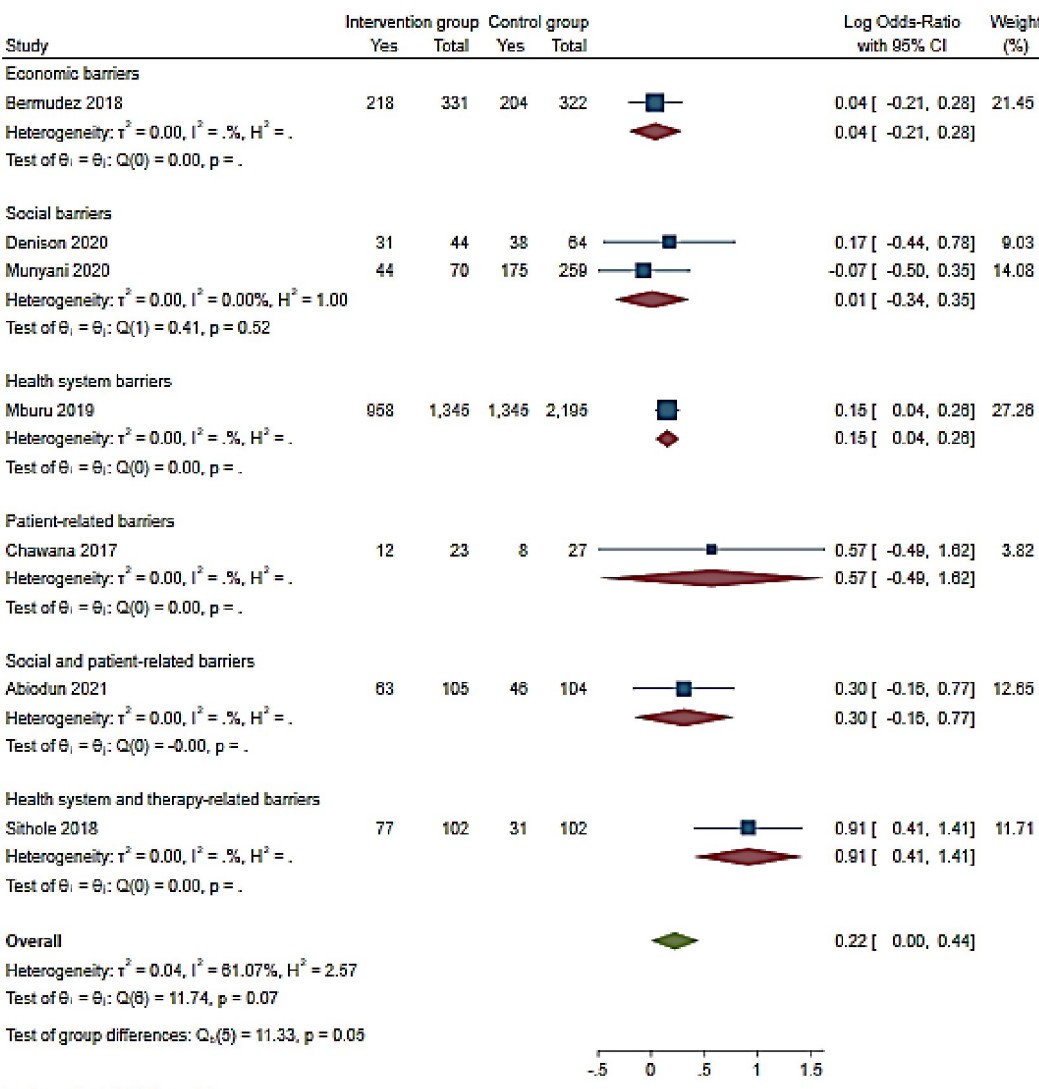

**Fig 6. Forest plot of viral load suppression and intervention reporting ART adherence barriers among adolescents aged from 10 to 19 years.**

### 3.5. Meta regression

A meta-regression analysis was performed to examine the ART the relationship between ART adherence and age group. ART adherence statistically reduced in the age group of 10 to 16 and 10 to 18 with P = 0.002 and 0.005, respectively. In contrast, a borderline decreased of adherence was observed in the age group of 10–19 years, P = 0.056 (Table 6).

### 3.6. Publication bias

The visual inspection of the funnel plots including ART adherence, viral load suppression and unsuppressed viral load rates were asymmetric (Figs 10–12). However, the Egger's tests for small study effects and Begg's test for ART adherence, viral load suppression and unsuppressed viral load rates were not statistically significant with the p-values of 0.2737; 0.3938, and 0.6147, respectively. Based on the above, publication bias may be considered less likely.

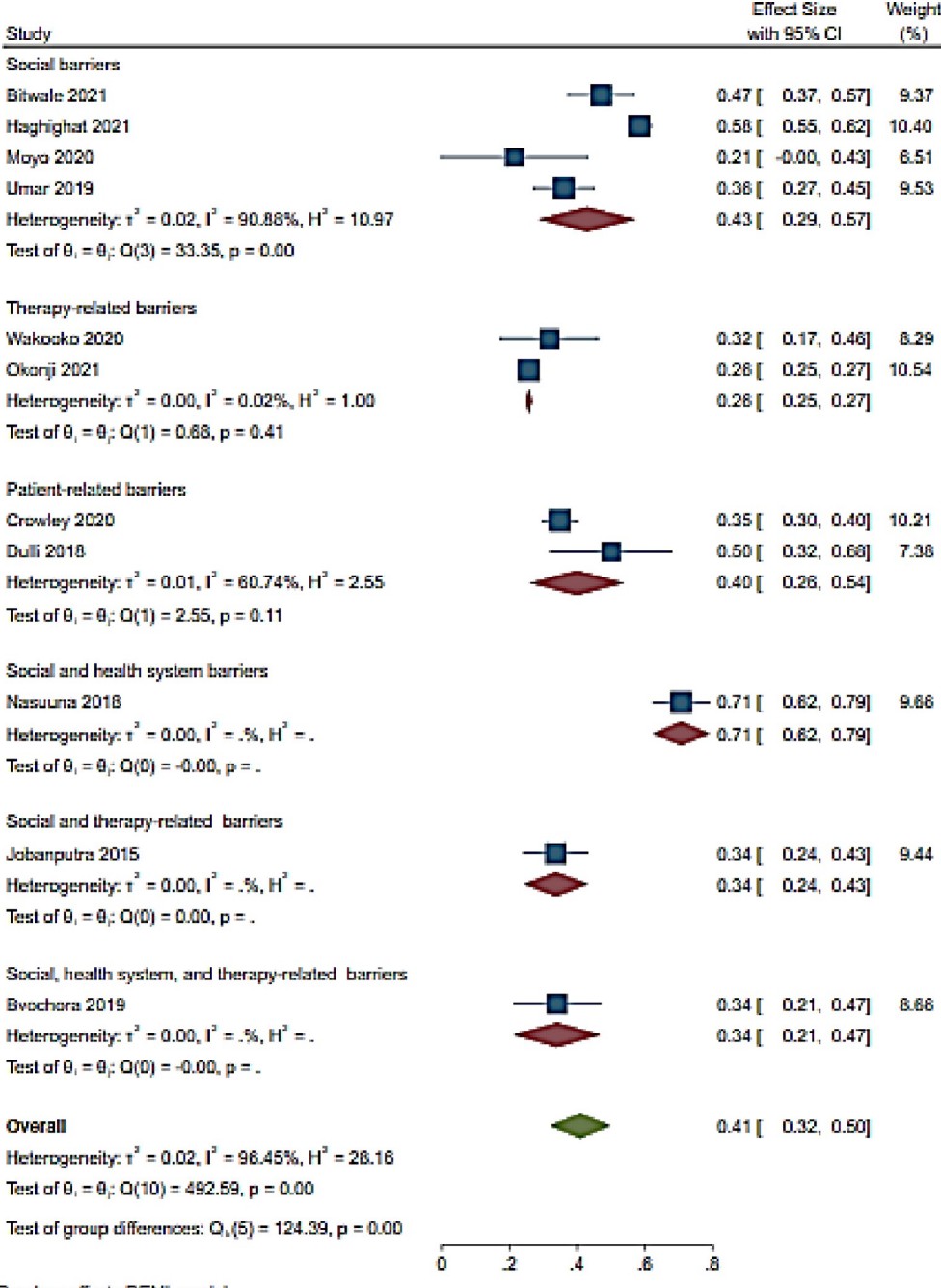

**Fig 7. Forest plot of unsuppressed viral load and ART adherence barriers among HIV infected adolescents aged from 10 to 19 years.**

### 3.7. Meta-synthesis

**3.7.1. Barriers of ART adherence.** *Social barriers*: Stigma and discrimination and lack of social support were the mostly identified social barriers to ART adherence among adolescents. Stigma and discrimination emanated from family members including parents and care-givers, community, friends, partners and school mates resulting to mental health effects such as

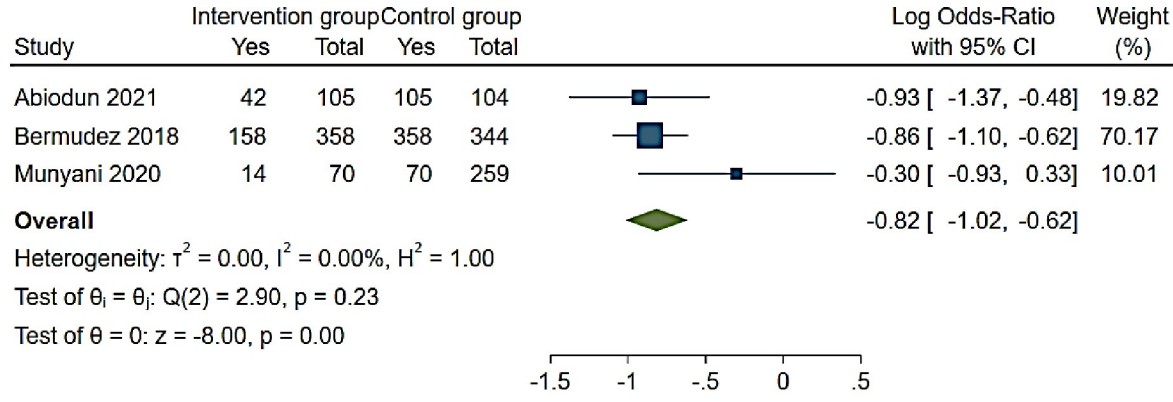

**Fig 8. Forest plot of ART adherence interventions and unsupressed viral load among adolescents aged from 10 to 19 years.**

depression and being suicidal [42, 45–47, 49, 50, 56, 57, 59, 100, 102, 103, 108, 109, 113]. While lack of support from caregivers and school mechanism resulted in adolescents being resentful and thus influenced their ART adherence patterns and resulting to hopelessness and suicidal thoughts too [49, 109, 113]. Lack of social support was evident among adolescents with a non-parent caregiver [42, 46, 49, 54, 55, 58, 106]. Within the school mechanism, studies reported the clash in medicine taking time, classes, exams or tests period, living environment for those in boarding schools, busy schedule and the appointment time at the health facilities as barriers to ART adherence [45, 49–52, 55, 56, 58, 59, 103, 104, 110].

*Economic barriers*: Poverty was reported to influence the ART adherence pattern among ALHIV. Poverty as explained in terms of lack of transport money to honor clinic schedules and lack of water and food to facilitate the correct uptake of medication resulted to adolescents missing clinic appointments and stopping medication [42, 51, 55, 58, 59, 104, 105].

*Heath system-related barriers*: Clinic or health facility factors were identified by adolescents as a barrier to ART adherence as a result of the location and distance to heath facility, waiting time at the facility, overcrowding, stock out, clinic protocols such as color coding, attitude of health-care workers, and rigid scheduling of appointments [40, 47, 49, 50, 52, 56, 104, 110]. These factors influenced the ART adherence patterns of the adolescents as they were not supportive to the needs of ALHIV resulting in loss to follow up and defaulting of medication [47, 110].

*Therapy-related barriers*: Regime, dosage frequency and side effects of medication were reported as the main barriers of ART adherence related to the treatment resulting to adolescents stopping medication [42, 46, 47, 50–52, 57]. Additionally, adolescents complained about the packaging of the pills as it was easy to identify what it is due to the labelling and also the bottles were large and made lots of noise risking unintended disclosure of their status and treatment [54, 57].

*Patient-related barriers*: The attitude of adolescents, adolescents' responsibility for treatment, and the *ART knowledge* were also a barrier to ART adherence as adolescents felt that being on ART posed restrictions on their life and caused them to miss opportunities their peers had [49, 54, 102]. Additionally, forgetfulness, tiredness, being busy, fatigue, being away from home were reported to also hinders adolescents' ART adherence [42, 46, 50, 52, 56, 58, 59, 102, 104, 106]. Adolescents stated that they also want to feel like others and just take pill holidays so they feel normal [106]. Lastly, adolescents described how not knowing what the medication, misconceptions and myths about ART made them less interested in adhering to taking it [102, 104].

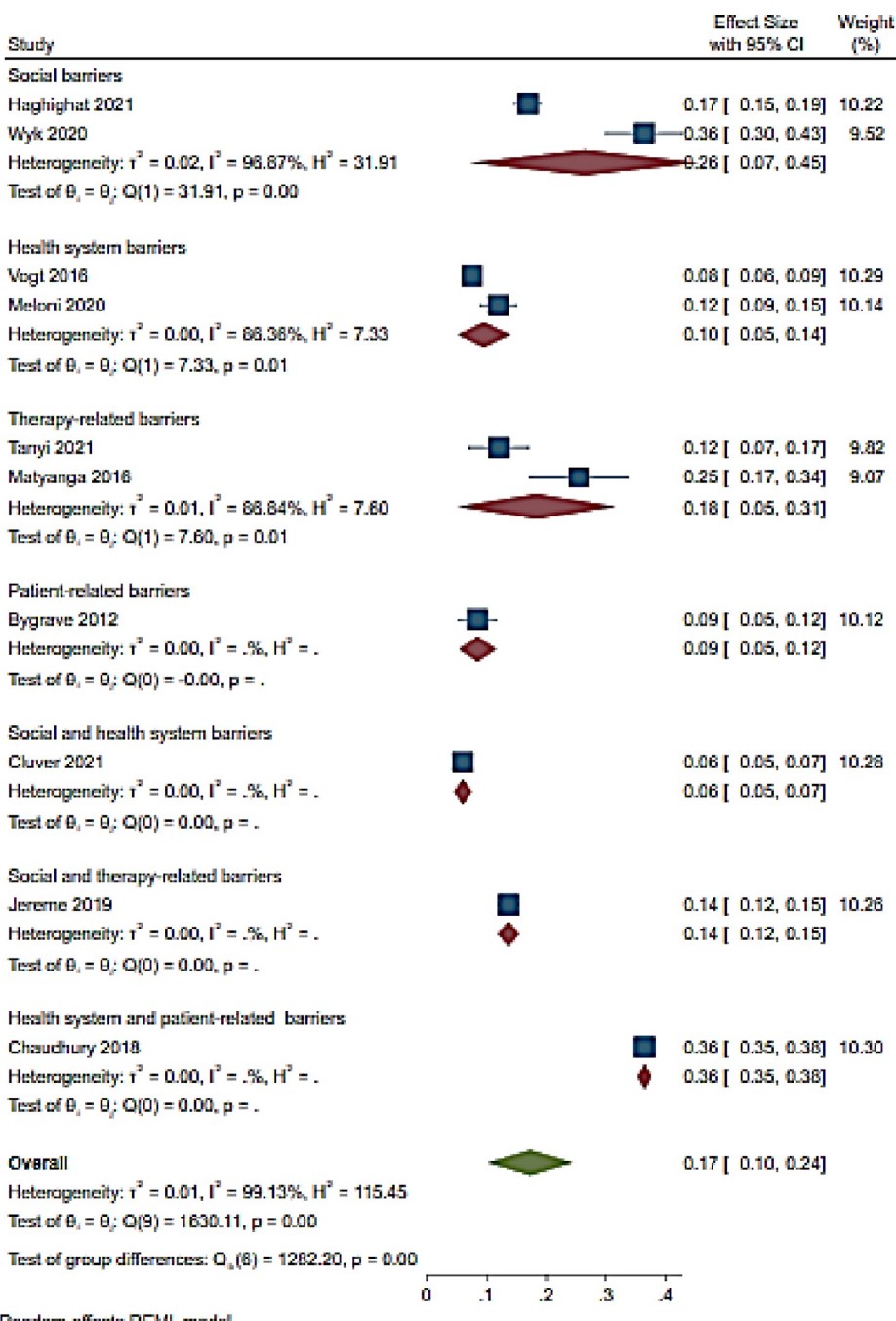

**Fig 9. Forest of LTFU and ART adherence barriers among adolescents aged from 10 to 19 years.**

*Cultural barriers*: Religion and culture had a strong influence on ART adherence among adolescents as they stopped taking their medication because they were prayed for, and were told to believe they were healed or believed that traditional medicine will heal them [55, 104].

**3.7.2. Facilitators of ART adherence.** Adolescents identified three themes to be important in promoting adherence to ART. These were support, secrecy or confidentiality and counselling or education. The facilitators are presented in Table 5.

**Table 6. Meta-regression output examining age group and ART adherence.**

| Age | Coefficient | Lower limit | Upper limit | Standard errors | P-value |
|---|---|---|---|---|---|
| 10–16 years | -0.509 | -0.824 | -0.194 | 0.161 | 0.002 |
| 10–17 years | 0.024 | -0.297 | 0.344 | 0.164 | 0.885 |
| 10–19 years | -0.218 | -0.441 | 0.006 | 0.114 | 0.056 |
| 10–18 years | -0.459 | -0.777 | -0.140 | 0.162 | 0.005 |
| 11–18 years | 0.168 | -0.159 | 0.494 | 0.167 | 0.314 |
| 13–18 years | 0.149 | -0.096 | 0.394 | 0.125 | 0.233 |

*Social Support*: Multiple studies reported that support system emanated from peers, family members, school, health facility, financial and personal or interpersonal support [46–48, 50, 51, 53, 57, 59, 104, 108, 109, 114],. These were fundamentals in mitigating stigma, discrimination and isolation and thereby foster networks and connectiveness of adolescents and their peers while creating a support structure to encourage adolescents to adhere to their medication especially when they are fatigued [42, 45, 47, 49, 51, 53, 54, 56–59, 100, 102, 104, 106, 109, 113]. Lastly, financial support was reported to be critical in ensuring adolescents had access to and could adhere to medication through provision of transport to the clinic and food supply [47, 53, 102, 104].

*Secrecy or confidentiality*: Status disclosure and safe storage of medication were reported to positively influence ART adherence as they enabled adolescents to take charge of their lives through adhering to the treatment as they understand the benefits of the medication [45, 50, 52, 54, 100, 110].

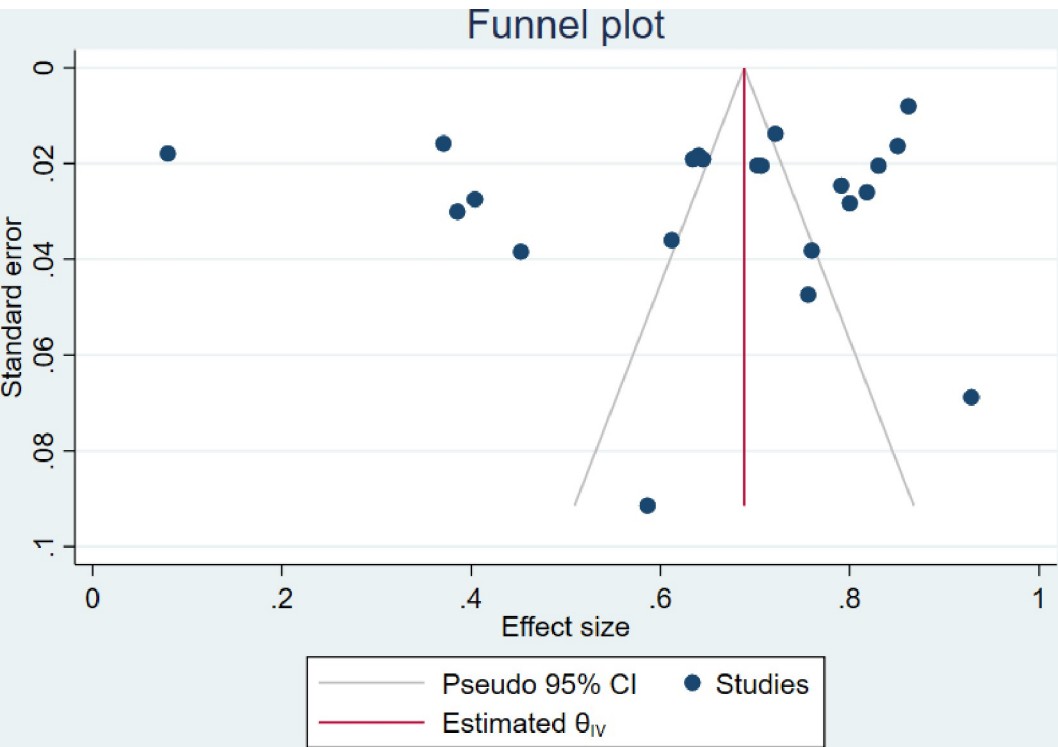

**Fig 10. Funnel plot of studies reporting ART adherence rate among adolescents aged from 10 to 19 years.**

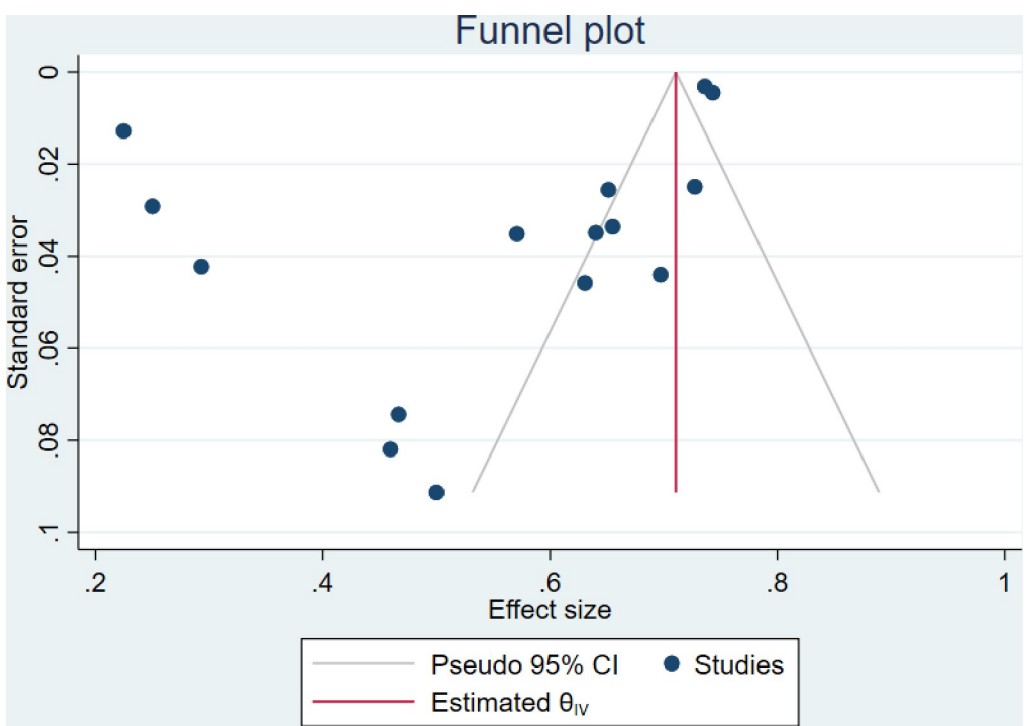

**Fig 11. Funnel plot of studies reporting viral load suppression rate among adolescents aged from 10 to 19 years.**

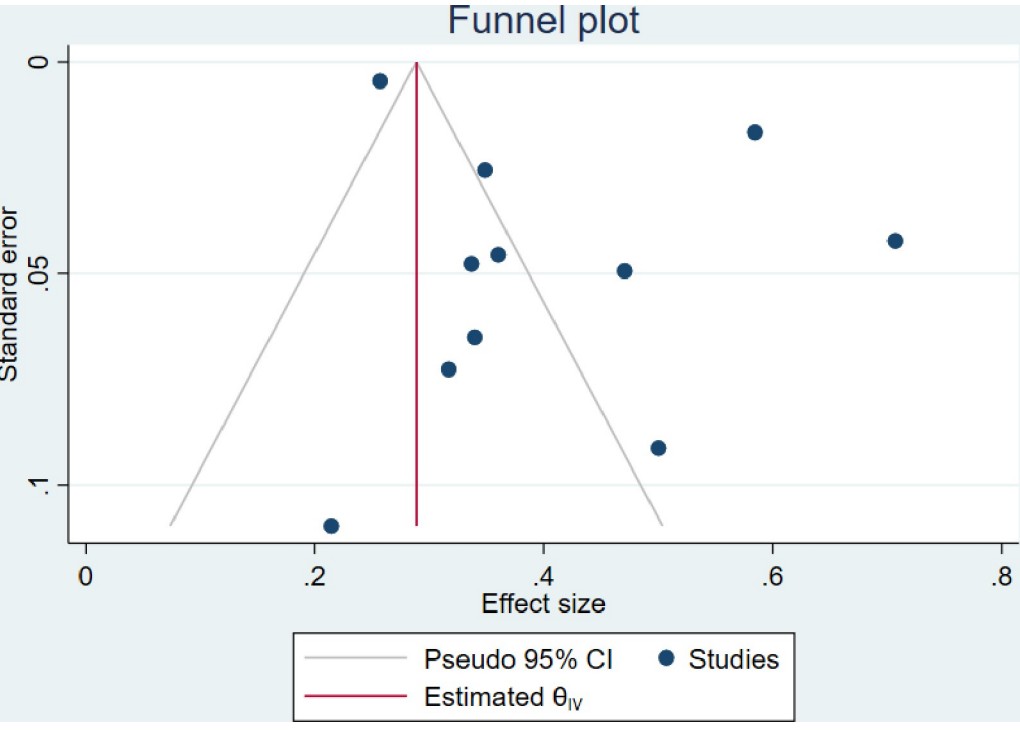

**Fig 12. Funnel plot of studies reporting unsuppressed viral load rate among adolescents aged from 10 to 19 years.**

*Counselling and ART education*: This gives adolescents an understanding of their status and importance of adhering while allaying their anxiety and discarding myths and misconceptions [42, 45, 49, 51, 100, 103, 106].

### 3.8. Overall results

An ART adherence rate of 65% (95% CI 56–74) was reported among adolescents in the SSA region. Viral load suppression rate was 55% (95% CI 46–64) while an unsuppressed viral load rate of 41% (95% CI 32–50) was reported with 17% (95% CI 10–24) loss to follow up rate. Six themes of barriers to ART adherence were identified among adolescents in SSA. These were social barriers, economic barriers, patient-related barriers, health systems-related and cultural barriers. Lastly, three main facilitating factors were identified being social support, secrecy or confidentiality and counselling, and ART education. The findings also revealed that the more barriers reported, the lower the adherence to ART among adolescents. According to the meta-regression, ART adherence was lower in the age group of 10 to 18 years. Poor ART adherence was caused by a composite of social and patient-related barriers. Adolescents had lower viral load suppression and a high rate of un-suppression due to social, health system, and patient-related barriers. Finally, health system and patient-related barriers reported a high rate of follow-up loss among adolescents. In comparison to the poor impact of interventions (SMS reminders, community adherence supporters, and teen clubs) on ART adherence, ART adherence facilitators have proven to be effective and acceptable among HIV-infected adolescents.

## 4. Discussion

The systematic review identified 66 studies which were included in the analysis which expanded on prior review on the same topic; ART adherence among adolescents [115–117]. The findings of our review showed that ART adherence was 65% (95% CI 56–74) and viral suppression was 55% (95% CI 46–64) which was in consistence with the 62% viral suppression used as a proxy measure for ART adherence among adolescent globally as reported in a systematic review including 53 countries [116]. There was a statistically positive association between ART adherence and adolescents age groups inclusive of the 10 to14 year age group. ART adherence was statistically poor among older adolescents which is comparable with studies reporting higher adherence among younger adolescents (10 to 14 years) compared to older adolescents (15 to 19 years). This could be explained by caregivers having higher levels of self-efficacy for medical care engagement and having good physician-caregiver relationships in early adolescence. In contrast, physical and social environments such as hostels for older adolescents' students, and stigma may play a substantial role in poor ART adherence in older adolescents. Furthermore, substance abuse may be a major factor in poor adherence in this age group. Studies have attributed the decrease in ART adherence among older adolescents to the diminishing support from care-givers during the transition from childhood to young adult and lack of second line ART especially in resource limited settings [42, 118–120]. However, the findings reported lower VLS rates compared to 2012 findings where VLS was at 84% in low- and middle-income countries yet was in consistent with findings of a systematic review reporting 59% in SSA [121]. Secondly, loss to follow up in our review was 17% (95% CI 10–24) which is lower to findings of a study conducted in 34 countries under the International epidemiology Database to Evaluate AIDS (IeDEA) where LTFU was reported to be 30% in 2018 [122]. Loss to follow up rate in this review indicate improvement when compared to studies conducted in 2007 in SSA where LTFU was 51% [123]. Additionally, our findings revealed that unsuppressed viral load is 41% in SSA countries which is higher than reported rates between 15.5% and 16.6 to 25.6% reported in Uganda and South Africa respectively [122, 123].

This however can be explained by findings of a systematic review conducted by Munyayi *et al* whereby different adolescent's focused intervention to improve ART adherence among countries reported differing viral load outcomes [124]. Lastly, interventions for ART adherence in included studies were primarily support oriented similar to findings to studies conducted in middle- and low-income countries even though our study indicated an increase in the number of interventions compared to previous studies [4, 13, 115]. However, our findings show that interventions such as SMS reminders, community adherence supporters, and teen clubs did not improve ART adherence in adolescents. In contrast, a study by Ba et al. concluded that teen-clubs, clinical psychological support and self-reinforcement strategies improved ART adherence among adolescents [118]. In a study conducted in low and middle-income countries, only home-based care was statistically associated with ART adherence while other interventions such as SMS reminders, economic support, enhanced counselling and multi-month refills were not statistically associated with ART adherence among adolescents [13]. In contrast, ART adherence facilitators such as social support, secrecy or confidentiality, counselling, and ART education were effective and acceptable. This could be explained by the fact that ART adherence facilitators considered a holistic approach to improving ART adherence [125, 126]. Rather than focusing on a specific adolescent aspect of ART adherence, as described in the interventions included in the meta-analysis, improving ART adherence may necessitate a more comprehensive approach. This is further demonstrated in the composite, which includes social barriers where ART adherence is low. A comprehensive approach to overcoming social barriers has shown that social support can improve ART adherence in adolescents. Although quantitative studies incorporating these adherence facilitators are required in Sub-Saharan Africa, a comprehensive approach should be considered to improve ART adherence in adolescents aged 10 to 19 years.

Stigma and discrimination, and lack of social support were mostly experienced and perceived by adolescents and had a negative impact on the treatment uptake and adherence. Stigma and discrimination were interrelated with other barriers (health system-related, patient-related, therapy-related, cultural, and economic barriers) perceived and experienced by adolescents and has a major contribution in the treatment uptake and adherence. Even though treatment uptake is an ultimate responsibility of adolescents, the clinic factors such as distance to the clinic, waiting time, health-workers perceived attitudes had a negative impact on the adherence rate of adolescents. The health system-related factors combined with the economic factors were also interrelated as without financial assistance, access to clinic or health facility is limited resulting to compromised ART adherence. Cultural barriers such as religion and use of traditional medicine facilitated by social barriers and therapy-related especially ART knowledge, side effects, WHO stage, stigma, and discrimination negatively impacted on the uptake and adherence of ART. Our study is consistent with studies that showed that personal beliefs, stigma, trust or satisfaction with health care workers, health service-related factors, and social support are main predictors for ART adherence [66, 110, 125–127]. Nevertheless, findings of this review indicate that these barriers can be addressed through proving social support, ART education and counselling of ALHIV. The identified facilitators are consistent with findings of a systematic review including studies conducted between 2004 and 2016 in SSA, which identified 29 facilitators which were mainly aimed at supporting adolescents through peer, care-giver, health-care workers, counseling, religion and beliefs of adolescents [115].

Our systematic review has strengths and limitations. Firstly, a key strength is that we conducted a mixed method systematic review which allows for comprehensive analysis of ART adherence among adolescents. Secondly, the review builds and improves on previously conducted studies in the region which included less studies in their analysis. Thirdly, even though

our review only included studies published in peer reviewed journals, being a mixed method systematic review justified the exclusion of unpublished literature. Fourthly, the use of a holistic quality appraisal process promoted the use of researchers' skills and judgement to assess value of contribution of each study. Therefore, the inclusion of all studies despite the quality assessment score allowed for contribution of each study to the knowledge pool thus valuable insight to the study subject. However, as a limitation of the review, only studies conducted in English were included yet there could be substantial evidence in missed studies conducted in other languages. Secondly, our review only included studies conducted from 2010 excluding previously conducted studies. Our review may therefore be missing other content of the subject matter which might still be applicable in the recent era and could have changed the perspective of the findings.

## 5. Conclusion

Multiple barriers to ART adherence have been reported among adolescents. These barriers negatively affect the uptake of ART and thereby the adherence to ART among adolescents. Nonetheless, several interventions aimed at improving ART adherence have been implemented in SSA. These interventions are aimed at creating a crucial support system for adolescents to address barriers to ART among adolescents. These barriers linked to lack of support systems either from caregivers, peers, the health system, and school environments. Despite the interventions, adolescents still present low ART adherence, viral load suppression and high lost to follow up and high unsuppressed viral load. This could be an indication of lack of adolescents input during the design of the intervention. A comprehensive approach may be the best way to improve ART adherence in adolescents aged 10 to 19 years. There is however a need for a study aimed at solely studying the influence of these interventions on the ART outcome.

## Supporting information

**S1 Table. Search strategy for the systematic literature review.**
(DOCX)

**S2 Table. Results of the quality assessment of included studies using the Mixed Method Appraisal Tool (MMAT).**
(DOCX)

**S3 Table. PRISMA 2020 checklist.**
(DOCX)

**S1 Fig. Forest plot of sub-group analysis of ART adherence among adolescents in sub-Saharan African countries.**
(TIF)

**S2 Fig. Forest plot of sub-group analysis of ART adherence and barriers to ART adherence among adolescents aged 10 to 19 years.**
(TIF)

## Author Contributions

**Conceptualization:** Londiwe D. Hlophe.

**Data curation:** Londiwe D. Hlophe, Jacques L. Tamuzi, Constance S. Shumba.

**Formal analysis:** Londiwe D. Hlophe, Jacques L. Tamuzi, Peter S. Nyasulu.

**Methodology:** Londiwe D. Hlophe, Jacques L. Tamuzi.

**Writing – original draft:** Londiwe D. Hlophe.

**Writing – review & editing:** Jacques L. Tamuzi, Constance S. Shumba, Peter S. Nyasulu.

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
