## [Decision Letter · Decision Letter 0]

6 Feb 2023

PONE-D-22-27523Barriers and facilitators to anti-retroviral therapy adherence among adolescents aged 10 to 19 years living with HIV in sub-Saharan Africa: A mixed-method systematic reviewPLOS ONE

Dear Dr. Nyasulu,

Thank you for submitting your manuscript to PLOS ONE. After careful consideration, we feel that it has merit but does not fully meet PLOS ONE’s publication criteria as it currently stands. Therefore, we invite you to submit a revised version of the manuscript that addresses the points raised during the review process.

 Journal Requirements: 

2. "Please identify your study as "Systematic review and meta-analysis" in the title of your study.

3. "We note that Figure 2,  in your submission contain [map/satellite] images which may be copyrighted. All PLOS content is published under the Creative Commons Attribution License (CC BY 4.0), which means that the manuscript, images, and Supporting Information files will be freely available online, and any third party is permitted to access, download, copy, distribute, and use these materials in any way, even commercially, with proper attribution. For these reasons, we cannot publish previously copyrighted maps or satellite images created using proprietary data, such as Google software (Google Maps, Street View, and Earth). For more information, see our copyright guidelines: http://journals.plos.org/plosone/s/licenses-and-copyright.

1. You may seek permission from the original copyright holder of Figure(s) [#] to publish the content specifically under the CC BY 4.0 license.  

Natural Earth (public domain): " ext-link-type="uri" xlink:type="simple">http://www.naturalearthdata.com/"

 Reviewers' comments:

Reviewer's Responses to Questions

**Comments to the Author**

1. Does the manuscript adhere to the experimental procedures and analyses described in the Registered Report Protocol?

If the manuscript reports any deviations from the planned experimental procedures and analyses, those must be reasonable and adequately justified.

Reviewer #1: Yes

Reviewer #2: Yes

2. If the manuscript reports exploratory analyses or experimental procedures not outlined in the original Registered Report Protocol, are these reasonable, justified and methodologically sound?

A Registered Report may include valid exploratory analyses not previously outlined in the Registered Report Protocol, as long as they are described as such.

Reviewer #1: Yes

Reviewer #2: Yes

3. Are the conclusions supported by the data and do they address the research question presented in the Registered Report Protocol?

The manuscript must describe a technically sound piece of scientific research with data that supports the conclusions. The conclusions must be drawn appropriately based on the research question(s) outlined in the Registered Report Protocol and on the data presented.

Reviewer #1: Yes

Reviewer #2: Yes

4. Have the authors made all data underlying the findings in their manuscript fully available?

Reviewer #1: Yes

Reviewer #2: Yes

5. Is the manuscript presented in an intelligible fashion and written in standard English?

Reviewer #1: Yes

Reviewer #2: Yes

6. Review Comments to the Author

Please use the space provided to explain your answers to the questions above. (Please upload your review as an attachment if it exceeds 20,000 characters)

Reviewer #1: Using the Assessment of Multiple Systematic Reviews (AMSTAR) Checklist, the authors have managed to include as much information on the checklist as possible (well done). While the authors have discussed how COVID-19 has affected HIV outcomes worldwide, it could be better if they add more literature on how this has affected the outcome of the included studies. Are there any differences in reported outcomes prior to COVID-19?

Reviewer #2: This is an important and timely study. The study is done according to the Preferred Reporting Items namely PRISMA statement, STROBE and ENTREQ recommendations.

There is great flow of the manuscript.

The methods are robust and well triangulated.

The findings are very clear.

7. PLOS authors have the option to publish the peer review history of their article (what does this mean?). If published, this will include your full peer review and any attached files.

**Do you want your identity to be public for this peer review?** For information about this choice, including consent withdrawal, please see our Privacy Policy.

Reviewer #1: No

Reviewer #2: **Yes: **Charles Masulani Mwale

A rebuttal letter that responds to each point raised by the academic editor and reviewer(s). You should upload this letter as a separate file labeled 'Response to Reviewers'.A marked-up copy of your manuscript that highlights changes made to the original version. You should upload this as a separate file labeled 'Revised Manuscript with Track Changes'.An unmarked version of your revised paper without tracked changes. You should upload this as a separate file labeled 'Manuscript'.If applicable, we recommend that you deposit your laboratory protocols in protocols.io to enhance the reproducibility of your results. Protocols.io assigns your protocol its own identifier (DOI) so that it can be cited independently in the future. For instructions see: https://journals.plos.org/plosone/s/submission-guidelines#loc-laboratory-protocols. Additionally, PLOS ONE offers an option for publishing peer-reviewed Lab Protocol articles, which describe protocols hosted on protocols.io. Read more information on sharing protocols at https://plos.org/protocols?utm_medium=editorial-emailutm_source=authorlettersutm_campaign=protocols.

We look forward to receiving your revised manuscript.

Kind regards,

Nyanyiwe Masingi Mbeye, Ph.D

Academic Editor

PLOS ONE
---

## [Author Response · Author response to Decision Letter 0]

27 Mar 2023

The authors have edited the manuscript and have ensured that it is now in line with the journal’s requirements.

2. "Please identify your study as "Systematic review and meta-analysis" in the title of your study.

The authors have edited the title and the study is now identified as a systematic review and meta-analysis.

3. "We note that Figure 2, in your submission contain [map/satellite] images which may be copyrighted. All PLOS content is published under the Creative Commons Attribution License (CC BY 4.0), which means that the manuscript, images, and Supporting Information files will be freely available online, and any third party is permitted to access, download, copy, distribute, and use these materials in any way, even commercially, with proper attribution. For these reasons, we cannot publish previously copyrighted maps or satellite images created using proprietary data, such as Google software (Google Maps, Street View, and Earth). For more information, see our copyright guidelines: http://journals.plos.org/plosone/s/licenses-and-copyright.

The map in our manuscript was created with Quantum Geographic information system (QGIS) software. We obtained African countries' shapefiles from a freely accessible database (https://www.naturalearthdata.com/downloads/). We geo-referenced the included studies in the shapefile based on their location as described in the table of included studies in our manuscript and joined the two files (shapefile and excel) to create the map. Based on the information provided, this map has not been published elsewhere and does not require permission

Reviewer #1 Using the Assessment of Multiple Systematic Reviews (AMSTAR) Checklist, the authors have managed to include as much information on the checklist as possible (well done). While the authors have discussed how COVID-19 has affected HIV outcomes worldwide, it could be better if they add more literature on how this has affected the outcome of the included studies. Are there any differences in reported outcomes prior to COVID-19?

While COVID-19 has been reported to have affected the HIV outcomes in general, our study could not be used to report on these effects or influences especially because some of the studies even though published post 2019/20, data was collected pre-COVID-19 and thus could be irrelevant in reporting the influence and impact of COVID-19 on adolescents ART adherence. However, authors have included more literature on the effects of COVID-19 on adolescents ART adherence.

Reviewer #2

1. Is there any reason why the authors did not consider viral suppression (another key outcome) in their search terms, while it is appearing in the methods section?

The authors have included the full search strategy as a supportive file (and viral load suppression and the other key ART outcomes have been included in the search strategy used for data collection.

2. On the results included, only one study was from Malawi studies which is not the same as in the tables. There may be a need to crosscheck for the other countries as well. 

The results on studied included are disintegrated by the two different studies types (qualitative and quantitative). Therefore, there is one study under qualitative (line 301, Burns 2020) and one under quantitative (line 316, Umar 2019). In total, there are 2 studies from Malawi (one under qualitative studies and the other under quantitative studies). The authors also cross checked the other studies. 

3. The discussion could have been broadened to address all the significant results presented in the results section. Its short.

Thank you, the authors have included more results in the discussion session including the meta-regression.

4. In the table for the studies included, from identification after removing the duplicates, to screening seems not to add up. 

The authors have explained the observed discrepancies in lines 294-296 and lines 307-310. Kindly note that in the mixed methods studies, in some studies only one component was included in our analysis as the other component did not meet the eligibility criteria of our study for instance study population not integrated by age group especially in studies on adolescents and children or adolescents and adults.

---

## [Decision Letter · Decision Letter 1]

2 May 2023

Barriers and facilitators to anti-retroviral therapy adherence among adolescents aged 10 to 19 years living with HIV in sub-Saharan Africa: A mixed-methods systematic review and meta-analysis

PONE-D-22-27523R1

Dear Dr. Nyasulu,

We’re pleased to inform you that your manuscript has been judged scientifically suitable for publication and will be formally accepted for publication once it meets all outstanding technical requirements.

Kind regards,

I. Marion Sumari-de Boer, Ph.D

Academic Editor

PLOS ONE

Additional Editor Comments (optional):

Reviewers' comments:

Reviewer's Responses to Questions

**Comments to the Author**

1. If the authors have adequately addressed your comments raised in a previous round of review and you feel that this manuscript is now acceptable for publication, you may indicate that here to bypass the “Comments to the Author” section, enter your conflict of interest statement in the “Confidential to Editor” section, and submit your "Accept" recommendation.

Reviewer #1: All comments have been addressed

Reviewer #2: All comments have been addressed

2. Is the manuscript technically sound, and do the data support the conclusions?

Reviewer #1: Yes

Reviewer #2: Yes

3. Has the statistical analysis been performed appropriately and rigorously? 

Reviewer #1: Yes

Reviewer #2: Yes

4. Have the authors made all data underlying the findings in their manuscript fully available?

Reviewer #1: Yes

Reviewer #2: Yes

5. Is the manuscript presented in an intelligible fashion and written in standard English?

Reviewer #1: Yes

Reviewer #2: Yes

6. Review Comments to the Author

Reviewer #1: None, the author have addressed all the comments that I raised in the first review. I think this is good to go and should be published

Reviewer #2: My coments have been adressed

7. PLOS authors have the option to publish the peer review history of their article (what does this mean?). If published, this will include your full peer review and any attached files.

Reviewer #1: **Yes: **Moses Banda Aron

Reviewer #2: **Yes: **Charles Masulani-Mwale

---

## [Editor Report · Acceptance letter]

5 May 2023

PONE-D-22-27523R1 

Barriers and facilitators to anti-retroviral therapy adherence among adolescents aged 10 to 19 years living with HIV in sub-Saharan Africa: A mixed-methods systematic review and meta-analysis 

Dear Dr. Nyasulu:

I'm pleased to inform you that your manuscript has been deemed suitable for publication in PLOS ONE. Congratulations! Your manuscript is now with our production department. 

Kind regards, 

on behalf of

Dr. I. Marion Sumari-de Boer 

Academic Editor

PLOS ONE